# Connexin hemichannels with prostaglandin release in anabolic function of bone to mechanical loading

Dezhi Zhao[1,2], Manuel A Riquelme[1], Teja Guda[3], Chao Tu[1,4], Huiyun Xu[2], Sumin Gu[1], Jean X Jiang[1]*

[1]Department of Biochemistry and Structural Biology, University of Texas Health Science Center at San Antonio, San Antonio, United States; [2]School of Life Sciences, Northwestern Polytechnical University, Xian, China; [3]Department of Biomedical Engineering and Chemical Engineering, University of Texas at San Antonio, San Antonio, United States; [4]Department of Orthopedics, The Second Xiangya Hospital, Central South University, Changsha, China

**Abstract** Mechanical stimulation, such as physical exercise, is essential for bone formation and health. Here, we demonstrate the critical role of osteocytic Cx43 hemichannels in anabolic function of bone in response to mechanical loading. Two transgenic mouse models, R76W and Δ130–136, expressing dominant-negative Cx43 mutants in osteocytes were adopted. Mechanical loading of tibial bone increased cortical bone mass and mechanical properties in wild-type and gap junction-impaired R76W mice through increased $PGE_2$, endosteal osteoblast activity, and decreased sclerostin. These anabolic responses were impeded in gap junction/hemichannel-impaired Δ130–136 mice and accompanied by increased endosteal osteoclast activity. Specific inhibition of Cx43 hemichannels by Cx43(M1) antibody suppressed $PGE_2$ secretion and impeded loading-induced endosteal osteoblast activity, bone formation and anabolic gene expression. $PGE_2$ administration rescued the osteogenic response to mechanical loading impeded by impaired hemichannels. Together, osteocytic Cx43 hemichannels could be a potential new therapeutic target for treating bone loss and osteoporosis.

*For correspondence:
jiangj@uthscsa.edu

Competing interest: The authors declare that no competing interests exist.

## Editor's evaluation

This study documents a key role for osteocyte-derived Cx43 hemichannels in the anabolic response of bone to mechanical stimulation, using transgenic mouse models and a Cx43 (M1) antibody. The studies shed new light on the molecular mechanisms that transduce mechanical loading into new bone formation, and identify Cx43 as an actionable target.

## Introduction

Bone as a mechanosensitive tissue is adaptive to mechanical stimuli, which are essential for bone homeostasis, formation, and remodeling (*Bonewald, 2011*). Reduced mechanical stimulation leads to bone loss and elevated risk of fracture (*Lang et al., 2004*), while enhanced mechanical stimulation, such as physical exercise, has positive, anabolic impacts on bone tissue, even following a prolonged cessation of stimulation (*Erlandson et al., 2012*; *Warden et al., 2007*). The osteocytes embedded in the bone mineral matrix comprise over 90–95% of all bone cells and are thought to be a major mechanoreceptor in the adult skeleton (*Bonewald, 2011*). Osteocytes detect the mechanical loading-induced alterations of the bone matrix microenvironment and translate them into biological responses

to regulate osteoblast and osteoclast activity on the bone surface (**Bonewald, 2011**; **Stenson et al., 1993**).

Connexin (Cx)-forming gap junctions and hemichannels permit small molecules ($\leq$ 1 kDa) to pass through the cellular membrane, such as prostaglandin $E_2$ ($PGE_2$) and ATP (**Loiselle et al., 2013**). Among Cx family members, Cx43 is the predominant Cx subtype expressed in osteocytes (**Civitelli, 2008**). Cx43 gap junctions allow cell-cell communication between osteocytes or between osteocytes and other bone cell types (**Ishihara et al., 2008**), and mechanical stimuli increase communication between two adjacent cells through gap junctions (**Alford et al., 2003**; **Cheng et al., 2001b**). However, osteocytic Cx43 gap junctions are only active at the tips of osteocyte dendritic processes and remain open even without mechanical stimulation (**Cusato et al., 2006**). In contrast, Cx43 hemichannels, which mediate the communication between the intracellular and the extracellular microenvironment, are highly responsive to mechanical stimulation in osteocytes (**Cherian et al., 2005**; **Jiang and Cherian, 2003**). Our previous studies have shown that in vitro mechanical stimulation, through fluid flow shear stress (FFSS), increases cell surface expression of Cx43 hemichannels (**Cherian et al., 2005**; **Jiang and Cherian, 2003**; **Siller-Jackson et al., 2008**), and opens Cx43 hemichannels, leading to the release of anabolic factor, $PGE_2$ in osteocytes (**Cherian et al., 2005**; **Siller-Jackson et al., 2008**). Activation of integrins and PI3K-Akt signaling by FFSS plays an essential role in activating Cx43 hemichannels (**Batra et al., 2012**; **Batra et al., 2014**). $PGE_2$ released by Cx43 hemichannels acts in an autocrine/paracrine manner to promote gap junction communication through transcriptional regulation of Cx43 (**Xia et al., 2010**) and blocks glucocorticoid-induced osteocyte apoptosis (**Kitase et al., 2010**). The opening of Cx43 hemichannels by FFSS also triggers the release of ATP by a protein kinase C-mediated pathway in osteocytes (**Genetos et al., 2007**). Extracellular $PGE_2$ accumulation caused by continuous FFSS exerts a negative feedback, leading to hemichannel closure (**Riquelme et al., 2015**). However, the biological role of osteocytic Cx43 hemichannels in the anabolic function of mechanical loading has remained largely elusive.

Several bone cell-type-specific Cx43 conditional knockout (cKO) mouse models have been reported. Deletion of Cx43 from osteoblasts and osteocytes driven by the 2.3 kb *Col1α1* promoter (*Col1α1-Cre; Gja1$^{-/flx}$*) attenuated tibial endosteal response to non-physiological mechanical loading, induced by four-point (**Grimston et al., 2006**) or three-point tibial bending (**Grimston et al., 2008**). However, deletion of Cx43 in osteochondroprogenitors driven by the *Twist2* promoter (*Twist2-cre; Gja1$^{-/flx}$*) (**Grimston et al., 2012**) or in osteoblasts/osteocytes driven by the *Bglap2* promoter (*Bglap2-Cre; Gja1$^{flx/flx}$*) (**Zhang et al., 2011**) showed an enhanced tibial periosteal response to tibial axial compression (**Grimston et al., 2012**) or tibial cantilever bending (**Zhang et al., 2011**). Similarly, deletion of Cx43 in osteocytes driven by an 8 kb dentin matrix protein 1 (*Dmp1*) promoter (*Dmp1-Cre; Gja1$^{flx/flx}$*) showed enhanced β-catenin levels and correspondingly increased periosteal response to ulna compression (**Bivi et al., 2013**). Interestingly, endosteal bone formation decreased more in *Twist2-Cre; Gja1$^{-/flx}$* mice (**Grimston et al., 2012**), but did not change in *Dmp1-Cre; Gja1$^{flx/flx}$* mice (**Bivi et al., 2013**) during mechanical loading. Together, these findings suggest that Cx43 plays a distinct role in the adaptive response to bone loading. However, since Cx43 forms both gap junctions and hemichannels, it has remained largely elusive whether the responses in knockout models could be attributed to either or both types of Cx43-forming channels. Here, we dissect the distinctive roles of Cx43 gap junctions and hemichannels using two transgenic mouse models that overexpress dominant-negative Cx43 mutants primarily in osteocytes with the 10 kb *Dmp1* promoter. The R76W transgenic mouse inhibits gap junctions with enhanced hemichannel function, whereas Δ130–136 inhibits both gap junctions and hemichannels (**Xu et al., 2015**). To further delineate the role of osteocytic Cx43 hemichannels under mechanical loading in vivo, a monoclonal Cx43(M1) antibody that specifically blocks Cx43 hemichannels was developed. In this study, we unveil a novel physiological role of Cx43 hemichannels in osteocytes and their release of $PGE_2$ in mediating anabolic function of the bone in response to mechanical loading.

## Results

### Impairment of Cx43 hemichannels attenuate anabolic responses of tibial bone to mechanical loading

In this study, we used two transgenic mouse models to distinguish the roles of osteocytic Cx43-gap junction channels and hemichannels in osteocytes in bone response to mechanical loading. We injected EB dye into mouse tail veins to determine the activity of Cx43 hemichannels in WT and transgenic mice in response to axial tibial loading. Bone tissue sections around the tibial midshaft region showed that tibial loading increased EB dye uptake in the osteocytes of WT and R76W mice, but not in the osteocytes of Δ130–136 mice (*Figure 1—figure supplement 2a,b*).

We subjected WT and transgenic mice with similar body weights (*Figure 1—figure supplement 3a*) to a 2-week cyclic tibial loading regime. μCT analyses of tibial metaphyseal trabecular bone showed that loading increased bone volume fractions (BV/TV) in WT and R76W mice (*Figure 1a*). In contrast, compared to contralateral, unloaded controls, tibial loading of Δ130–136 mice exhibited a significant reduction of trabecular number (Tb.N) and bone mineral density (BMD), as well as increased trabecular separation (Tb.Sp) during mechanical loading (*Figure 1b, c and e*). However, compared to contralateral, unloaded tibias, trabecular thickness (Tb.Th) was increased in loaded tibias of WT and two transgenic mice (*Figure 1d*). There was no change of structural model index (SMI), indicating that loading did not affect the shape of trabecular bone (*Figure 1f*). Representative μCT images of trabecular bone are shown in *Figure 1g*.

Similar attenuation of anabolic responses to tibial loading was also observed in cortical bone. μCT analysis was conducted at the midshaft cortical bone (50% site). Loading increased bone area (B.Ar), bone area fraction (B.Ar/T.Ar), and cortical thickness (Ct.Th) in WT and R76W mice (*Figure 2b, c and e*). Although T.Ar was increased by mechanical loading in Δ130–136 (*Figure 2a*), enlarged bone marrow area (M.Ar) (*Figure 2d*) attenuated the ratio of B.Ar/T.Ar (*Figure 2c*). The increased Ct.Th. due to tibial loading was not observed in Δ130–136 mice (*Figure 2e*). Interestingly, the loading caused a decrease of BMD in R76W mice (*Figure 2f*). Torsional strength, predicted by polar moment of inertia (pMOI), was increased as a result of mechanical loading in WT and R76W, but not in Δ130–136 mice (*Figure 2g*). Representative images of cortical bone are shown in *Figure 2h*. Together these data suggested that osteocytic Cx43 hemichannels, not gap junctions, play an important role in anabolic responses of both trabecular and cortical bones to mechanical loading.

### Cx43 hemichannels mediate endosteal osteogenic responses to mechanical loading

Dynamic histomorphometric analyses were performed to evaluate periosteal and endosteal bone formation in response to tibial loading. Loading caused a significant increase of endosteal MAR, MS/BS, and BRF/BS compared to contralateral tibias in WT and R76W, but such an increase was not observed in Δ130–136 (*Figure 3a–d*). The decreased endosteal bone formation may partially account for the enlarged bone marrow. Contrary to the endosteal surface, Δ130–136 mice showed a statistically significant osteogenic response on the periosteal surface compared to WT and R76W mice, manifesting a threefold increase in MAR, MS/BS, and BRF/BS during loading (*Figure 3e–h*). Together, these data suggested that impaired osteocytic Cx43 hemichannels in Δ130–136 mice attenuated endosteal bone formation, and enhanced periosteal bone formation upon mechanical loading.

### Impaired Cx43 hemichannels inhibit the loading-induced PGE$_2$ secretion and osteoblast activity, and promote osteoclast activity

Cx43 hemichannels mediate PGE$_2$ release from osteocytes induced by FFSS in vitro (*Cherian et al., 2005*), and extracellular PGE$_2$ is reported to play a key role in the anabolic response to mechanical loading of bone tissue (*Jee et al., 1985*; *Thorsen et al., 1996*). We measured PGE$_2$ levels in the tibial bone diaphysis and found PGE$_2$ levels in mechanically loaded tibias were significantly increased in WT and R76W mice compared to those in contralateral, non-loaded tibias (*Figure 4a*). However, loading had minimal effect on PGE$_2$ levels in Δ130–136 mice. Immunohistochemical staining showed that the expression of cyclooxygenase-2 (COX-2), a key enzyme that catalyzes the conversion of arachidonic acid to prostaglandins, was significantly increased in the osteocytes of loaded tibias of WT and R76W mice compared to contralateral, unloaded tibias (*Figure 4b and c* and *Figure 4—figure supplement*

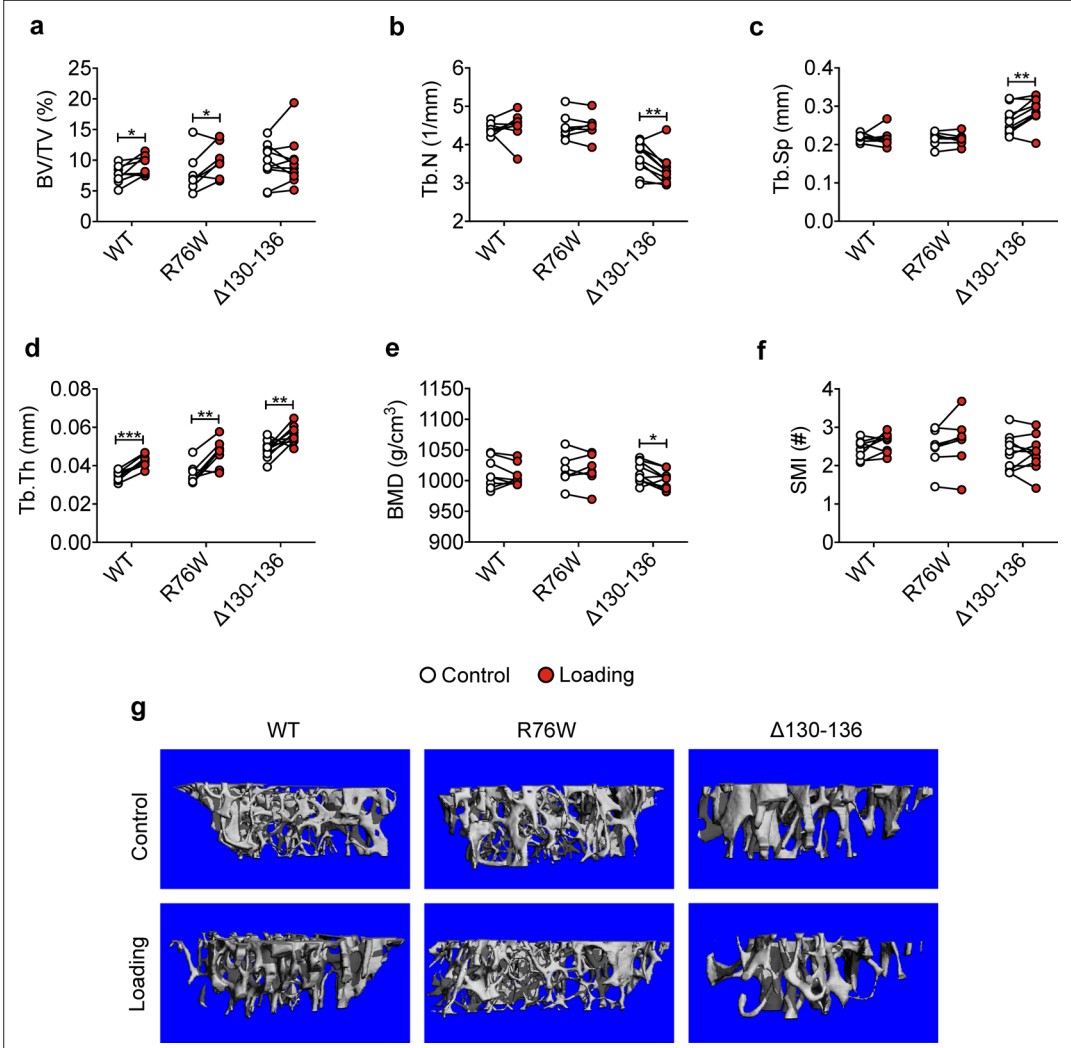

**Figure 1.** Attenuation or reversal of anabolic responses to mechanical loading in tibial metaphyseal trabecular bone of Δ130–136 mice. μCT was used to assess metaphyseal trabecular bone of WT, R76W, and Δ130–136 mice; (**a**) bone volume fraction, (**b**) trabecular number, (**c**) trabecular separation, (**d**) trabecular thickness, (**e**) bone mineral density, and (**f**) structure model index. n = 7–10/group. (**g**) Representative 3D models of the metaphyseal trabecular bone for all groups. Data are expressed as mean ± SD. *, p < 0.05; **, p < 0.01; ***, p < 0.001. Statistical analysis was performed using paired t-test for loaded and contralateral, unloaded tibias within each genotype.

The online version of this article includes the following source data and figure supplement(s) for figure 1:

**Source data 1.** Trabecular micro-CT data of transgenic and wild-type mice.

**Figure supplement 1.** Experimental setup for in vivo axial loading.

**Figure supplement 1—source data 1.** Raw data of compliances for *Figure 1—figure supplement 1c*.

**Figure supplement 2.** Hemichannel opening is inhibited in Δ130–136 mice.

**Figure supplement 2—source data 1.** Raw data of dye uptake for *Figure 1—figure supplement 2b*.

**Figure supplement 3.** Body weights of transgenic mice.

**Figure supplement 3—source data 1.** Raw data of body weight for *Figure 1—figure supplement 3*.

---

*1a*). However, the increase of COX-2 in loaded tibias was not observed in Δ130–136 mice. SOST-positive osteocytes decreased significantly in WT and R76W in response to tibial loading, while such decrease in loaded tibias was absent in Δ130–136 mice (*Figure 4d and e* and *Figure 4—figure supplement 1b*). A similar reduction at the mRNA level of the bone was also found in WT and R76W, but absent in Δ130–136 mice (*Figure 4f*). Since SOST, a Wnt receptor antagonist, is a potent inhibitor

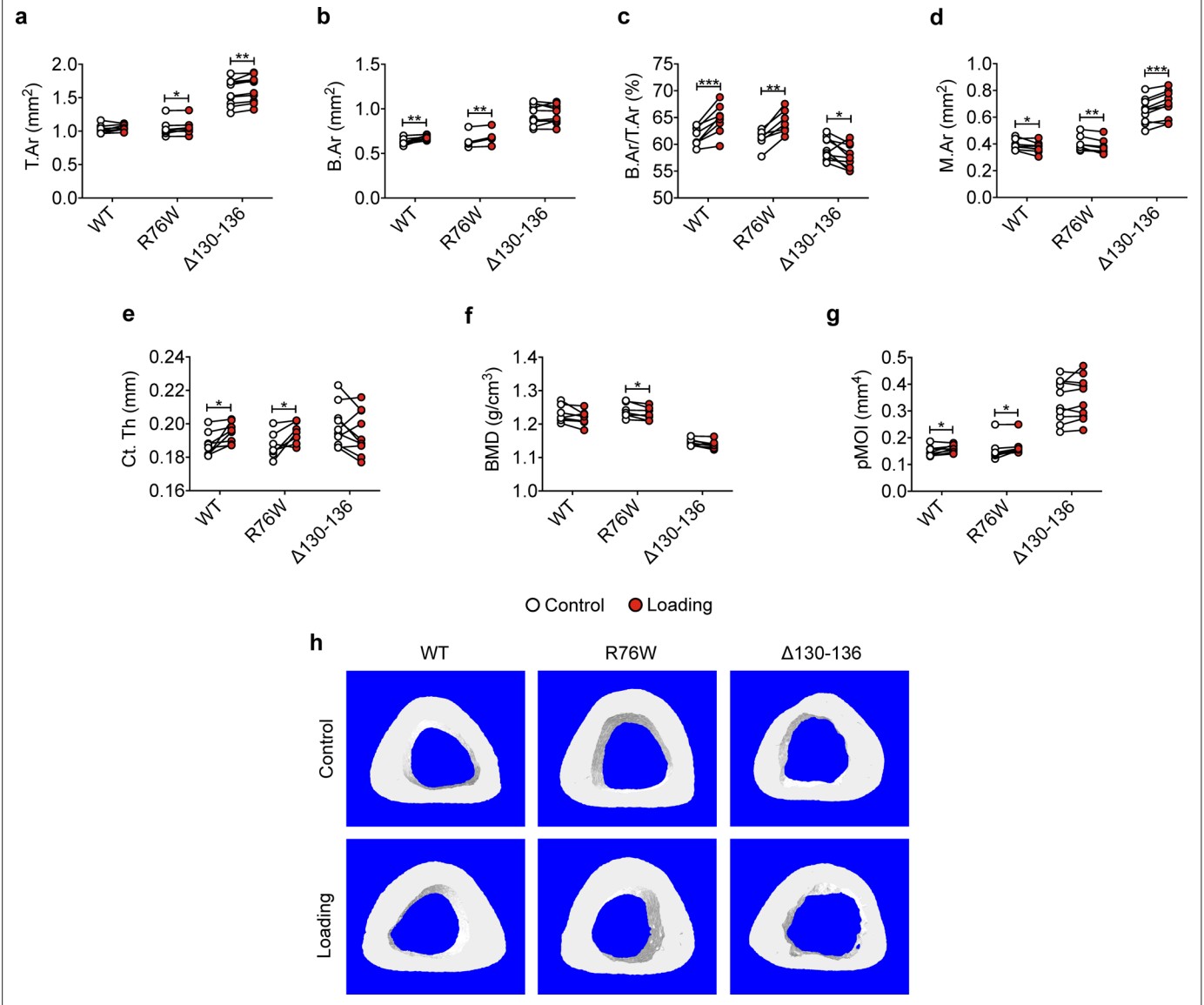

**Figure 2.** Attenuation or reversal of anabolic responses to mechanical loading in midshaft cortical bone of Δ130–136 mice. μCT was used to assess tibial midshaft cortical bone (50% site) of WT, R76W, and Δ130–136 mice; (**a**) total area, (**b**) bone area, (**c**) bone area fraction, (**d**) bone marrow area, (**e**) cortical thickness, (**f**) bone mineral density, and (**g**) polar moment of inertia. n = 7–10/group. (**h**) Representative 3D models of the tibial midshaft cortical bone for all groups. Data are expressed as mean ± SD. *, p < 0.05; **, p < 0.01; **, p < 0.001. Statistical analysis was performed using paired t-test for loaded and contralateral, unloaded tibias within each genotype.

The online version of this article includes the following source data for figure 2:

**Source data 1.** Cortical micro-CT data of transgenic and wild-type mice.

of osteoblastic activity, we examined osteoblasts on the endosteal surface. WT and R76W mice exhibited an increase of osteoblast numbers on the endosteal surface; in contrast, this increase was absent in Δ130–136 mice (***Figure 4g and h***). Moreover, the levels of mRNA of osteoblastic markers *Runx2* and *Bglap2* were increased in the bone of loaded WT and R76W, but absent in Δ130–136 mice (***Figure 4i and j***). The mRNA expression of the osteocytic marker *Dmp1* in the bone of Δ130–136 mice showed a similar trend of reduction compared to that of control and R76W mice (***Figure 4k***). Contrary to osteoblasts, osteoclast number on the endosteal bone surface was significantly increased in loaded tibias in Δ130–136 mice (***Figure 4l–n***). The data suggested that osteocytic Cx43 hemichannels influence PGE$_2$ secretion, key bone marker expression, and osteoblastic and osteoclastic activities on endosteal surfaces in response to mechanical loading.

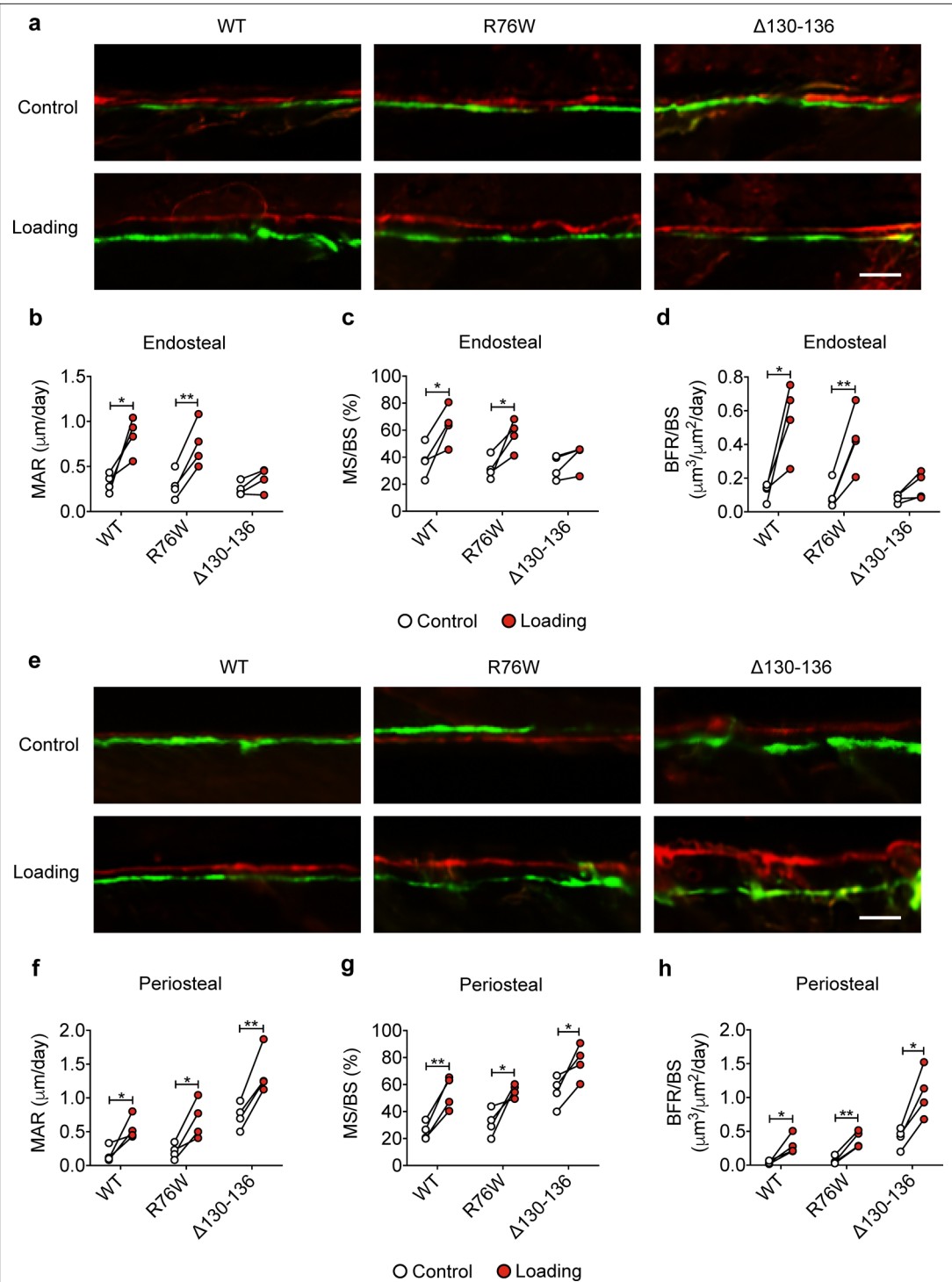

**Figure 3.** Reduced midshaft endosteal osteogenic responses to mechanical loading in Δ130–136 mice. Dynamic histomorphometric analyses were performed on the tibial midshaft cortical endosteal and periosteal surfaces after 2 weeks of tibial loading of WT, R76W, and Δ130–136 mice. Representative images of calcein (green) alizarin (red) double labeling on (**a**) endosteal and (**e**) periosteal surface. Scale bar: 50 μm. Mineral apposition rate (MAR) (**b, f**), mineralizing surface/bone surface (MS/BS) (**c, g**), and bone formation rate (BFR/BS) (**d, h**) were assessed for endosteal (**b–d**) and periosteal (**f–h**) surfaces n = 4/group. Data are expressed as mean ± SD. [*], p < 0.05; [**], p < 0.01. Statistical analysis was performed using paired t-test for loaded and contralateral, unloaded tibias within each genotype.

The online version of this article includes the following source data for figure 3:

**Source data 1.** Raw data of periosteal and endosteal bone formation of transgenic and wild-type mice.

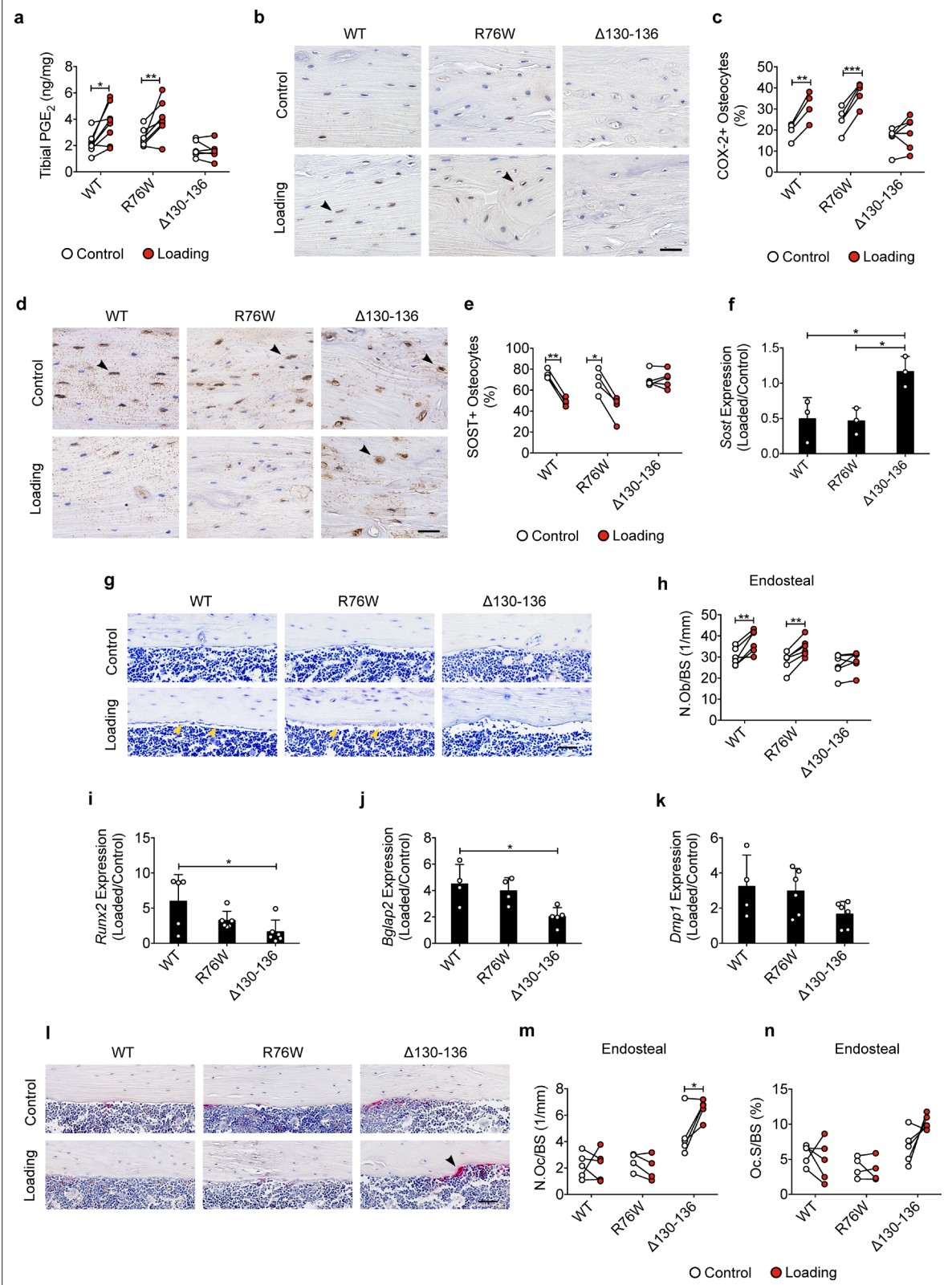

**Figure 4.** Inhibition of the loading-induced PGE$_2$ secretion and osteoblast activity, and promotion of osteoclast activity in Δ130–136 mice. (**a**) ELISA analysis of PGE$_2$ in bone marrow-flushed tibial diaphysis after 5 days of tibial loading, in WT, R76W, and Δ130–136 mice. n = 6–8/group. (**b, c**) Representative images and quantitative analysis of COX-2-postive osteocytes (black arrows) in the tibial midshaft cortical bone after 2 weeks of loading in WT, R76W, and Δ130–136 mice. Scale bar: 30 μm. n = 4–6/group. (**d, e**) Representative images and quantitative analysis of the SOST-positive

*Figure 4 continued on next page*

*Figure 4 continued*

osteocytes (black arrows) in tibial midshaft cortical bone after 2 weeks of tibial loading in WT, R76W, and Δ130–136 mice. Scale bar: 30 μm. n = 4–5/ group. (**f**) Gene expression of *Sost* in bone marrow-flushed tibial diaphysis of WT, R76W, and Δ130–136 mice. n = 5/group. (**g and h**) Toluidine blue staining was used to determine the number of endosteal osteoblasts (yellow arrows) on tibial midshaft cortical bone in WT, R76W, and Δ130–136 mice after 2 weeks of loading. Scale bar: 30 μm; n = 6/group. mRNA expression of osteoblast markers, *Runx2* (**i**), *Bglap2* (**j**) and *Dmp1* (**k**) in bone marrow-flushed tibial diaphysis of WT, R76W, and Δ130–136 mice. n = 4–5/group. (**l**) Representative images of tibial midshaft endosteal surface stained for TRAP (black arrows). Scale bar: 30 μm. (**m and n**) Histomorphometric quantitation of osteoclasts per bone perimeter (**m**) and osteoclast surface per bone perimeter (**n**) (n = 4–5/group). All quantitative data are expressed as mean ± SD. *, p < 0.05; **, p < 0.01; ***, p < 0.001. Statistical analysis was performed using paired t-test for loaded and contralateral, unloaded tibias within each genotype.

The online version of this article includes the following source data and figure supplement(s) for figure 4:

**Source data 1.** Raw data of PGE$_2$ level for *Figure 4a*.

**Source data 2.** Raw data of immumohistochemical, TRAP and toluidine blue staining of transgenic and wild-type mice.

**Source data 3.** Raw data of RT-qPCR of transgenic and wild-type mice.

**Figure supplement 1.** Bone marker protein expression in WT, R76W, and Δ130–136 mice.

**Figure supplement 1—source data 1.** Raw data of COX-2 and SOST quantification for *Figure 4—figure supplement 1a,b*.

## Cx43 hemichannel-blocking antibody impairs the anabolic effects of mechanical loading on trabecular and cortical bones

We have developed a polyclonal antibody, Cx43(E2), that targets an extracellular loop domain of Cx43 and specifically blocks osteocytic Cx43 hemichannels (*Siller-Jackson et al., 2008*). We recently developed a specific mouse monoclonal blocking antibody Cx43(M1) to investigate the roles of mouse hemichannels in vivo. Similar to Cx43(E2), Cx43(M1) had a strong reactivity with Cx43 (*Figure 5— figure supplement 1a*). Gap junction channels and hemichannels were assayed using dye coupling (*Figure 5—figure supplement 1b*) and dye uptake assays (*Figure 5—figure supplement 1c*), respectively. Both Cx43(E2) and Cx43(M1) had minimal effects on gap junction channels as indicated by comparable levels of dye transfer with red-orange AM dye in MLO-Y4 cells (*Figure 5—figure supplement 1b*). Conversely, FFSS-induced hemichannel opening, as determined by EtBr uptake, was inhibited by Cx43(E2) and Cx43(M1) antibodies in MLO-Y4 cells (*Figure 5—figure supplement 1c*). The extent of inhibition is comparable between Cx43(E2) and Cx43(M1) antibodies. To ensure antibody delivery to osteocytes, we labeled tibial bone sections with a rhodamine-conjugated anti-mouse secondary antibody. Strong antibody signals were primarily detected in osteocytes in cortical bone for Cx43(M1)-injected mice, but not in IgG-injected ones (*Figure 5—figure supplement 1d*). Interestingly, low levels of Cx43(M1) were detected in trabecular bone (*Figure 5—figure supplement 1e*). The hemichannel opening detected by EB fluorescence was found only in loaded tibial bone, and this uptake was almost completely blocked by the Cx43(M1) antibody (*Figure 5—figure supplement 1f,g*). Hence, these studies established the feasibility of using Cx43(M1) antibody to assess the role of Cx43 hemichannels in vivo using the tibial loading model.

WT mice with similar body weight (*Figure 5—figure supplement 2a*) were randomly allocated to Cx43(M1) or vehicle groups. A slight decline in body weight was found during the first week of the study, but was stabilized by the second week (*Figure 5—figure supplement 2a*). The antibody had a negligible effect on body weight (*Figure 5—figure supplement 2a*). μCT analysis of tibial metaphyseal trabecular bone showed that Cx43(M1) treatment significantly abated the loading-induced increase of Tb.N and decrease of Tb.Sp (*Figure 5b and c*) as compared to the vehicle-treated group, while Cx43(M1) did not manifest significant differences in the Tb.Th, BV/TV, and BMD, in response to loading (*Figure 5d and f*). There was no change of SMI in both Cx43(M1) or vehicle groups (*Figure 5g*). Representative images of trabecular bone are shown in *Figure 5a*.

Cx43(M1) treatment attenuated the anabolic response to mechanical loading in midshaft cortical bone. The increase of B.Ar, B.Ar/T.Ar, and Ct.Th by tibial loading was attenuated in the Cx43(M1)-treated group (*Figure 5j, k and m*), and consequently, the increase of pMOI was also attenuated in this group (*Figure 5o*). Similar to Δ130–136 mice, larger M.Ar and lower B.Ar/T.Ar ratios were found in loaded tibias of the Cx43(M1)-treated group compared to contralateral, unloaded tibias (*Figure 5k and l*). Although Cx43(M1) further increased T.Ar by mechanical loading (*Figure 5i*), enlarged M.Ar (*Figure 5l*) significantly reduced the ratio of B.Ar/T.Ar (*Figure 5k*). Thus, Ct.Th did not increase (*Figure 5m*). However, BMD was not changed by mechanical loading (*Figure 5n*). Three-point bending

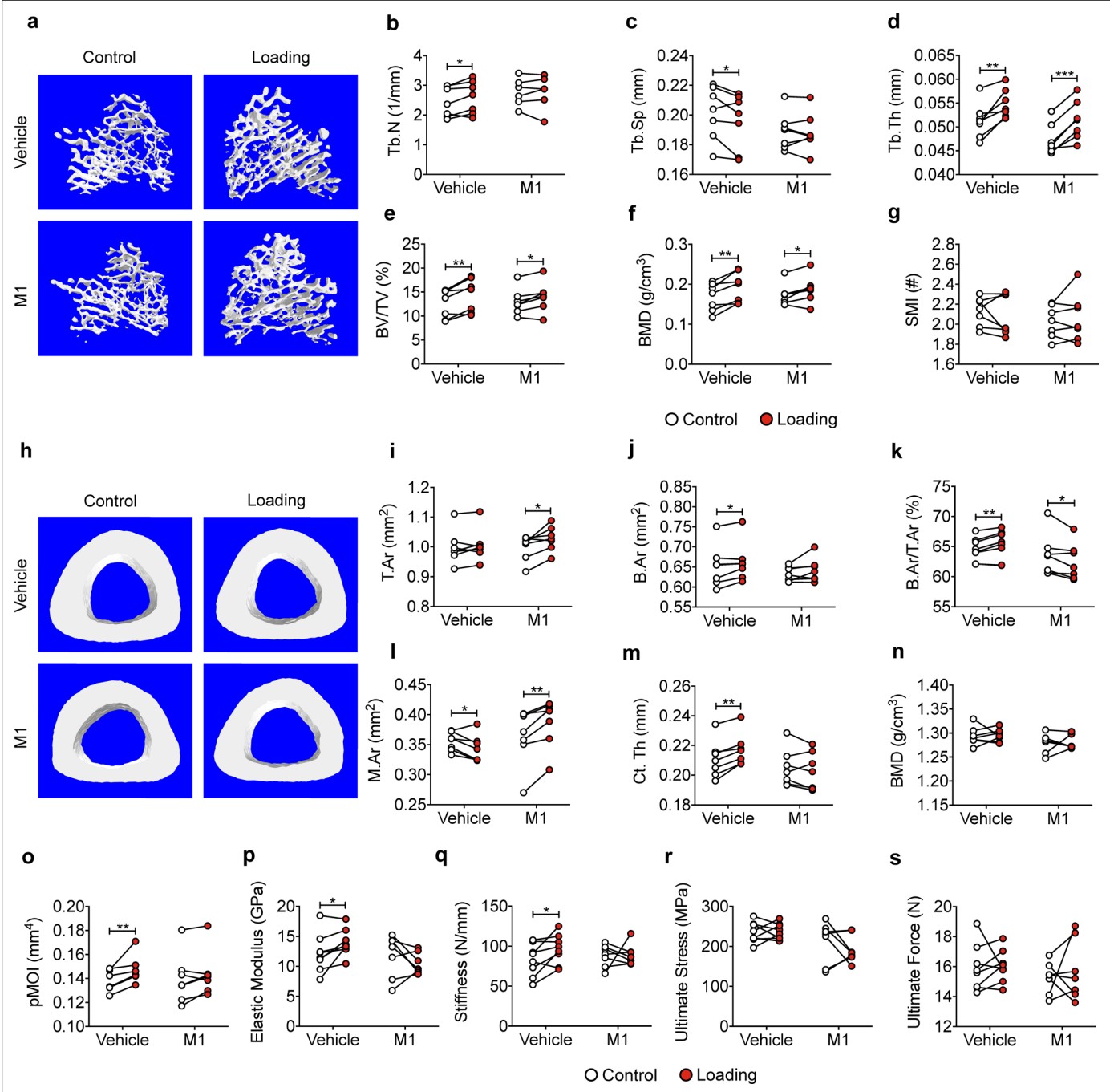

**Figure 5.** Inhibition of Cx43 hemichannels by Cx43(M1) antibody impairs anabolic effects of mechanical loading on trabecular and cortical bones. (**a**) Representative 3D models of the metaphyseal trabecular bone of vehicle and Cx43(M1)-treated mice. (**b–g**) µCT was used to assess structural parameters of trabecular bone; (**b**) trabecular number, (**c**) trabecular separation, (**d**) trabecular thickness, (**e**) bone volume fraction, (**f**) bone mineral density, and (**g**) structure model index in vehicle and Cx43 (**M1**)-treated mice. n = 7/group. (**H**) Representative 3D models of the tibial midshaft cortical bone (50% site) in vehicle and Cx43(M1)-treated mice. (**i–n**) µCT was used to assess structural parameters of cortical bone; (**i**) total area, (**j**) bone area, (**k**) bone area fraction, (**l**) bone marrow area, (**m**) cortical thickness, (**n**) bone mineral density and (**o**) polar moment of inertia in vehicle and Cx43(M1)-treated mice. n = 7/group. (**p–s**) The three-point bending assay was performed for tibial bone of vehicle and Cx43 (**M1**)-treated mice; (**p**) elastic modulus, (**q**) stiffness, (**r**) ultimate stress, and (**s**) ultimate force. n = 7–8/group. Data are expressed as mean ± SD. *, p < 0.05; **, p < 0.01; ***, p < 0.001. Statistical analysis was performed using paired t-test for loaded and contralateral within each treatment.

The online version of this article includes the following source data and figure supplement(s) for figure 5:

**Source data 1.** Micro-CT data of vehicle and Cx43(M1)-treated mice.

*Figure 5 continued on next page*

*Figure 5 continued*

**Source data 2.** Three-point bending data of vehicle and Cx43(M1)-treated mice.

**Figure supplement 1.** Monoclonal antibody of Cx43 inhibits hemichannel opening induced by mechanical stress in vitro and in vivo.

**Figure supplement 1—source data 1.** Raw data of dye uptake for *Figure 5—figure supplement 1c and g*.

**Figure supplement 1—source data 2.** Full scan of western blots for *Figure 5—figure supplement 1a*.

**Figure supplement 2.** Body weights of mice during 2 weeks of tibial loading.

**Figure supplement 2—source data 1.** Raw data of body weight for *Figure 5—figure supplement 2a*.

analyses revealed a significant increase of elastic modulus and stiffness only in the vehicle group (*Figure 5p and q*). Mechanical loading did not change ultimate stress and ultimate force in either vehicle or Cx43(M1)-treated group (*Figure 5r and s*). Representative images of cortical bone are shown in *Figure 5h*. These results are consistent with those obtained from Δ130–136 mice, suggesting the critical roles of Cx43 hemichannels in the anabolic effects of mechanical loading of cortical bone.

## Blocking Cx43 hemichannels by Cx43(M1) inhibits the load-induced increase in midshaft endosteal osteogenesis

Bone formation in response to tibial loading was evaluated in vehicle and Cx43(M1)-treated mice. The vehicle group exhibited increased endosteal MAR, MS/BS, and BRF/BS compared to contralateral, unloaded controls, whereas this response was absent in the Cx43(M1) group (*Figure 6a–d*). In contrast, loading increased bone formation in vehicle and Cx43(M1)-treated groups on the periosteal surface (*Figure 6e–h*). The results showed that impaired Cx43 hemichannels attenuated endosteal bone formation, but enhanced periosteal bone formation, induced by mechanical loading.

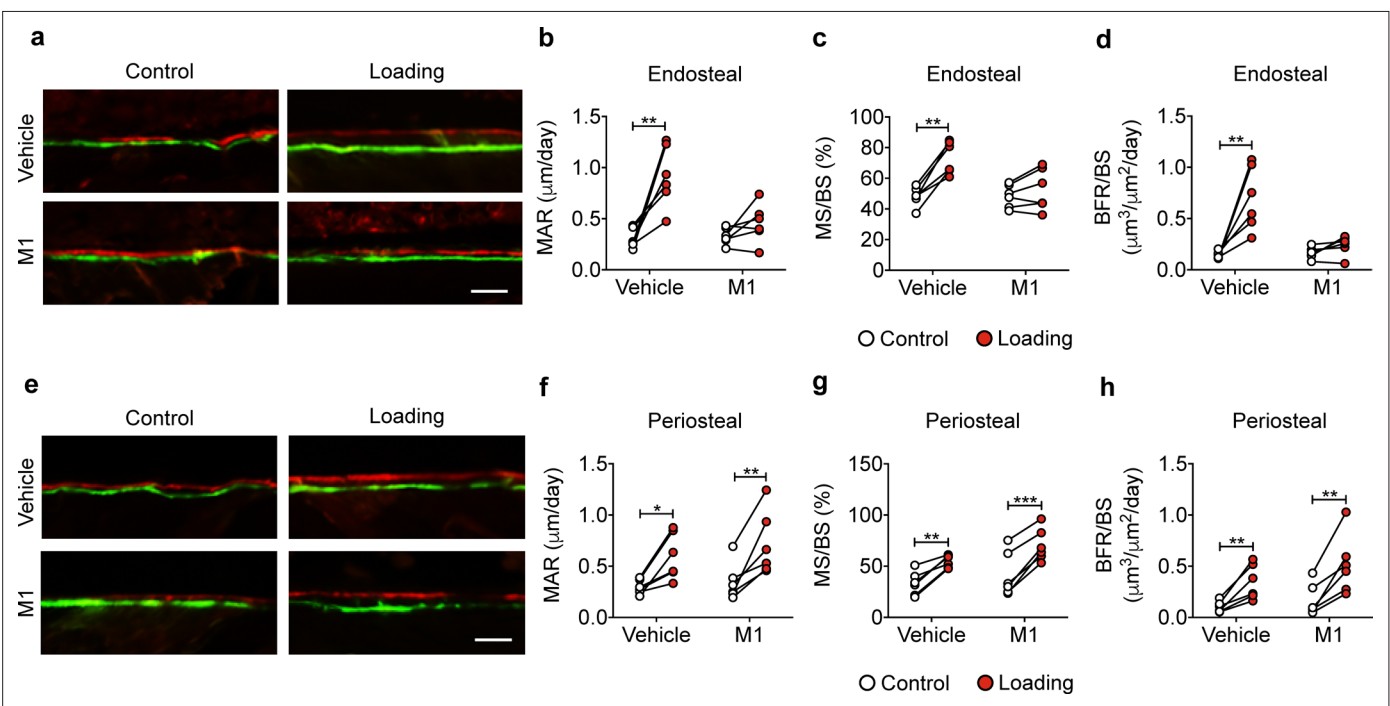

**Figure 6.** Cx43(M1) inhibits the load-induced increase in midshaft endosteal osteogenesis. Dynamic histomorphometric analyses were performed on the tibial midshaft cortical endosteal (**a–d**) and periosteal (**e–h**) surfaces after 2 weeks of loading in vehicle and Cx43(M1)-treated mice. (**a, e**) Representative images of calcein (green) alizarin (red) double labeling on (**a**) endosteal and (**e**) periosteal surface Scale bar: 50 μm. Mineral apposition rate (MAR), mineralizing surface/bone surface (MS/BS), and bone formation rate (BFR/BS) were assessed for (**b–d**) endosteal and (**f–h**) periosteal surfaces (n = 6/group). Data are expressed as mean ± SD. *, p < 0.05; **, p < 0.01; ***, p < 0.001. Statistical analysis was performed using paired t-test for loaded and contralateral within each treatment.

The online version of this article includes the following source data for figure 6:

**Source data 1.** Raw data of periosteal and endosteal bone formation of vehicle and Cx43(M1)-treated mice.

## Blocking Cx43 hemichannels by Cx43(M1) impedes the loading-induced increased PGE$_2$ secretion, bone marker expression, and endosteal osteoblastic activity, and decreased osteoclastic activity

Tibial loading significantly increased PGE$_2$ expression in tibial bone, and this increase was not observed with Cx43(M1) antibody treatment (*Figure 7a*). Immunohistochemical staining of tibial cortical bone showed a significant increase of COX-2 positive osteocytes by mechanical loading, and such increase was not detected in the Cx43(M1)-treated group (*Figure 7b and c* and *Figure 7—figure supplement 1a*). *Ptgs2* mRNA levels in bone detected by RT-qPCR exhibited close to a fivefold increase due to tibial loading, and this increase was not detected in Cx43(M1) treated mouse bone samples (*Figure 7d*). Loading caused a significant decrease of SOST-positive osteocytes in tibial bone in the vehicle group, and Cx43(M1) antibody abated the load-induced decrease of SOST-positive osteocytes (*Figure 7e and f* and *Figure 7—figure supplement 1b*). *Sost* mRNA in bone also showed a significant decrease due to loading in the vehicle group compared to the Cx43(M1)-treated group (*Figure 7g*). We next determined another mechanical response protein, β-catenin expression, in osteoblasts. Tibial loading resulted in a robust increase in endosteal β-catenin-positive osteoblasts and tibial *Ctnnb1* gene expression in the vehicle group; in contrast, such increase was absent in the Cx43(M1) group (*Figure 7h–j* and *Figure 7—figure supplement 1c*). Consistent with β-catenin expression, endosteal osteoblast number only increased in the vehicle group (*Figure 7k and l*). Moreover, the increase of gene expression of the osteoblastic marker, *Bglap2* was found in the vehicle group, but absent in the Cx43(M1) group (*Figure 7m*). Interestingly, increased osteoclast activity was also found in the Cx43(M1) group (*Figure 7k, n and o*). The results showed that under mechanical loading, inhibition of Cx43 hemichannels by Cx43(M1) antibody impedes the PGE$_2$ release and SOST decrease in osteocytes. This was associated with inhibited β-catenin and osteocalcin expression and osteoblast activity on the endosteal surface.

## PGE$_2$ rescues impeded osteogenic responses to mechanical loading by impaired Cx43 hemichannels

Intermittent PGE$_2$ treatment has been reported to increase both trabecular and cortical bone mass (*Jee et al., 1985*; *Tian et al., 2007*). To explore whether the attenuated anabolic function of bone to mechanical loading with Cx43(M1) is caused by inhibited PGE$_2$ released by Cx43 hemichannels, PGE$_2$ was IP injected into the vehicle control and Cx43(M1)-treated mice daily during the 2-week cyclic tibial loading. The mice in the control and treated groups had comparable body weights to minimize variations in tibial bone sizes before loading (*Figure 8—figure supplement 1a*). μCT analysis showed that there was no difference in trabecular morphometric parameters among the four groups after 2-week tibial loaidng (*Figure 8—figure supplement 2*). Contrary to trabecular bone, PGE$_2$ treatment impeded the significant reduction of B.Ar/T.Ar ratio, decreased M.Ar in Cx43(M1)-treated loaded tibias, although there were no significant differences in T.Ar, B.Ar, and BMD in all four groups (*Figure 8b–g*). Representative images of cortical bone are shown in *Figure 8a*. Interestingly, PGE$_2$ did not further enhance anabolic bone responses in control, loaded mice. Together, these results demonstrate that administration of PGE$_2$ significantly rescues impeded anabolic responses of cortical bone to mechanical loading as a result of Cx43 hemichannel inhibition.

## Discussion

In this study, we investigated the distinctive in vivo roles of two types of Cx43 channels in responses to mechanical loading using both transgenic mouse models and Cx43 hemichannel-blocking antibodies, and demonstrated the critical physiogical role of Cx43 hemichannels in mediating the anabolic, or bone forming, function of bone upon mechanical loading. Moreover, administration of PGE$_2$, a factor released by Cx43 hemichannels in response to mechanical stimulation, rescued the impeded anabolic effects on cortical bone by tibial loading because of impaired hemichannels.

We determined bone structural and biomechanical properties, new bone formation, and metabolism using an established bone mechanical loading model, axial tibial compression, in WT and two transgenic mouse models expressing Cx43 dominant negative mutants R76W and Δ130–136 in osteocytes. We observed that unlike WT and R76W mice, Δ130–136 mice, with inhibited osteocytic Cx43 hemichannels, attenuated anabolic bone responses to mechanical loading in both trabecular and

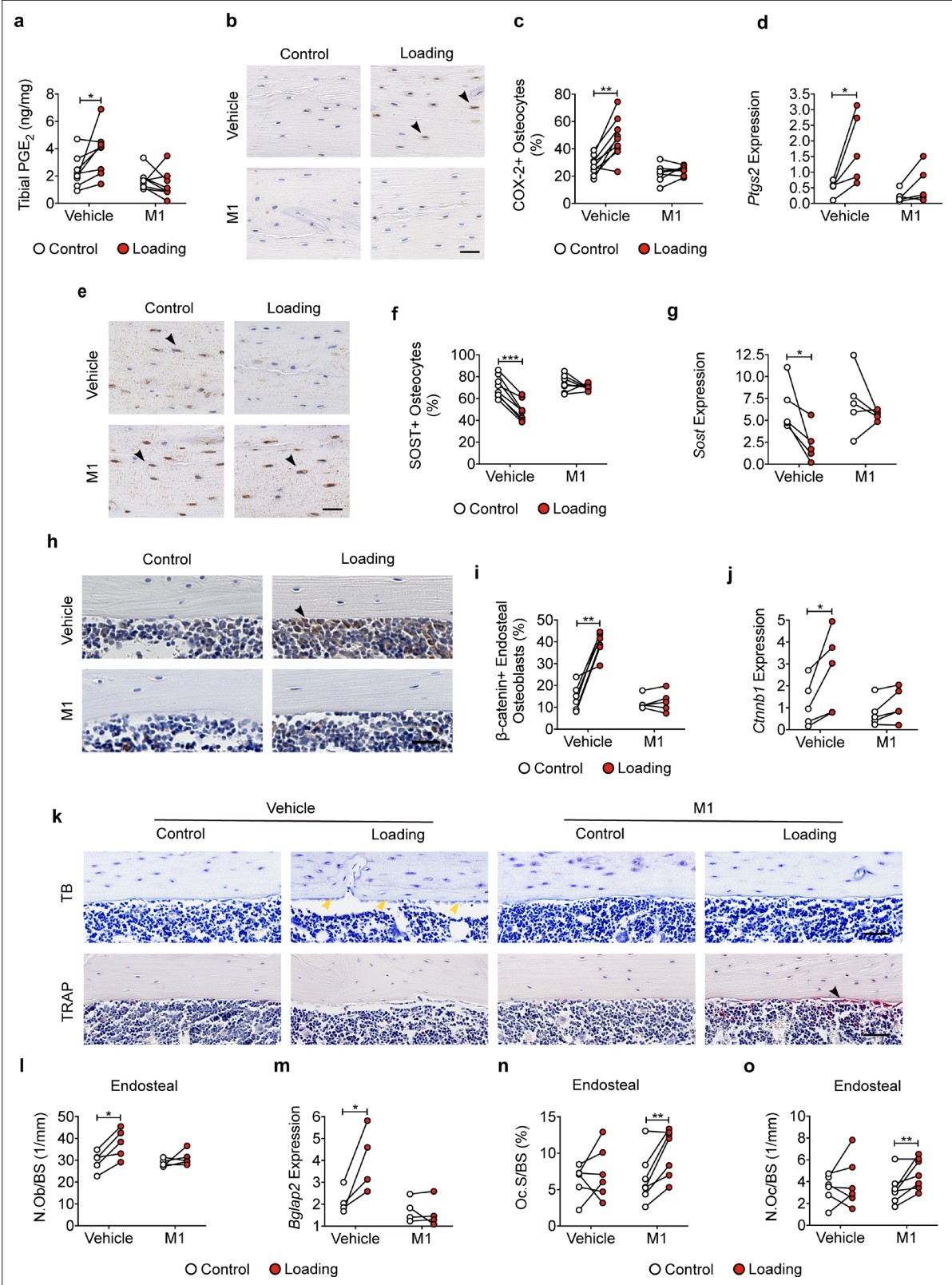

**Figure 7.** Cx43(M1) impedes the loading-induced increased PGE$_2$ secretion and osteoblast activity, and decreased osteoclast activity. (**a**) ELISA analysis of PGE$_2$ in bone marrow-flushed tibial diaphysis after 5 days of mechanical loading in vehicle and Cx43(M1)-treated mice (n = 8/group). (**b, c**) Representative images and quantitative analysis of COX-2-postive osteocytes (yellow arrows) in tibial midshaft cortical bone after 2 weeks of loading in vehicle and Cx43(M1)-treated mice. Scale bar: 30 μm. n = 8–9/group. (**d**) *Ptgs2* mRNA determined by RT-qPCR in bone marrow-flushed tibial diaphysis

*Figure 7 continued on next page*

*Figure 7 continued*

of vehicle and Cx43(M1)-treated mice. n = 4/group. (**e, f**) Representative images and quantitative analysis of the SOST-positive osteocytes (yellow arrows) in tibial midshaft cortical bone after 2 weeks of mechanical loading in vehicle and Cx43(M1)-treated mice. Scale bar: 30 μm (n = 8/group). (**g**) *Sost* mRNA determined by RT-qPCR from bone marrow-flushed tibial diaphysis of vehicle and Cx43(M1)-treated mice. n = 4/group. (**h, i**) Representative images and quantitative analysis of the β-catenin-positive periosteal cells (black arrows) on tibial midshaft endosteal surface after 2 weeks of loading in vehicle and Cx43(M1)-treated mice. Scale bar: 20 μm; n = 5–6/group. (**j**) *Ctnnb1* mRNA determined by RT-qPCR in bone marrow-flushed tibial diaphysis of vehicle and Cx43(M1)-treated mice. n = 4/group. (**k**) Representative images of tibial midshaft endosteal surface stained for toluidine blue (top panel) or TRAP (low panel). The yellow arrows indicate osteoblasts and the black arrows indicate the TRAP-positive osteoclasts. Scale bar: 30 μm. (**l**) Histomorphometric quantitation of osteoblast per bone perimeter (n = 5–7/group). (**m**) *Bglap2* mRNA determined by RT-qPCR in bone marrow-flushed tibial diaphysis of vehicle and Cx43(M1)-treated mice. n = 4/group. (**n, o**) Histomorphometric quantitation of osteoclast per bone perimeter (**n**) and osteoclast surface per bone perimeter (**o**) (n = 5–7/group). Data are expressed as mean ± SD. *, p < 0.05; **, p < 0.01; ***, p < 0.001. Statistical analysis was performed using paired t-test for loaded and contralateral within each treatment.

The online version of this article includes the following source data and figure supplement(s) for figure 7:

**Source data 1.** Raw data of PGE$_2$ level for *Figure 7a*.

**Source data 2.** Raw data of immumohistochemical, TRAP and toluidine blue staining of vehicle and Cx43(M1)-treated mice.

**Source data 3.** Raw data of RT-qPCR of vehicle and Cx43(M1)-treated mice.

**Figure supplement 1.** Bone marker protein expression in vehicle- and Cx43(M1)-treated mice.

**Figure supplement 1—source data 1.** Raw data of COX-2, SOST, and β-catenin quantification for *Figure 7—figure supplement 1a-c*.

cortical bones. In cortical bones of WT and R76W mice, the bone formation in both periosteal and endosteal surface were increased by tibial loading and this increase is correlated with the increase of bone area fractions and cortical thickness. In Δ 130–136 mice, only bone formation on the periosteal surface increased, but not on the endosteal surface. In addition, we observed the increased osteoclast number in endosteal surface. The net effect in Δ 130–136 is a decreased bone area fraction, cortical thickness and an enlarged bone marrow area. Due to impaired osteocytic gap junction channels and hemichannels in Δ130–136 mice, but only impaired gap junction channels in R76W mice, we postulated that osteocytic Cx43 hemichannels, not gap junctions in osteocytes, are likely to play a predominant role in anabolic bone response to mechanical loading.

The role of Cx43 hemichannels was further validated by the Cx43(M1) antibody, a potent monoclonal antibody that effectively inhibits osteocytic Cx43 hemichannels both in vitro and in vivo. Remarkably, treatment with Cx43(M1) only twice in a span of 2 weeks significantly attenuated anabolic effects to mechanical loading, with greater effects in cortical bones. Interestingly, Cx43(M1) treatment not only attenuated, but even reversed anabolic effects in cortical bone, similar to Δ130–136 mice. A similar impediment of the rate of bone formation and mineral apposition to tibial loading was observed on the endosteal surface with Cx43(M1) treatment. Previous studies have shown that cortical bone modeling/remodeling is more pronounced at the endosteal surface (*Birkhold et al., 2017*), and mature bones respond to mechanical loading through changes on endosteal surfaces (*Bass et al., 2002*). Similar findings were also noted in Cx43 cKO mouse models; mice lacking Cx43 in osteoblasts and osteocytes showed an attenuated increase in endosteal bone formation during four-point or three-point tibial bending (*Grimston et al., 2008*; *Grimston et al., 2006*). Mice lacking Cx43 in osteochondroprogenitors showed a greater extent of decrease in endosteal formation during tibial compression loading (*Grimston et al., 2012*). In our study, notably, endosteal MAR and BFR/BS were not responsive to tibial loading in Δ130–136 mice and the Cx43(M1) group, suggesting not only increased osteoblastic activity, but also that increased osteoblast number was impeded by the impairment of Cx43 hemichannels. Moreover, histomorphometric analysis further confirmed a lack of response in osteoblast number and marker genes in Δ130–136 mice and Cx43(M1)-treated mice. The difference between Cx43 cKO and our transgenic models and Cx43(M1) treatment could be caused by aberrant, compensatory effect of other pathways as a result of Cx43 deletion. Thus, our results suggest that axial compression loading promotes osteoblast recruitment and differentiation on the endosteal surface, an anabolic effect likely mediated by mechanosensitive Cx43 hemichannels.

Interestingly, contrary to our hypothesis, axial load increased periosteal bone formation and total tissue area on the tibial midshaft in Δ130–136 mice. This observation was also reported in cKO mouse models with Cx43 deletion in osteoblasts and osteocytes under tibial axial compression (*Grimston et al., 2012*) and tibial cantilever bending (*Zhang et al., 2011*). Similarly, deletion of Cx43 in osteocytes also showed an enhanced periosteal response to ulnar compression (*Bivi et al., 2013*). The

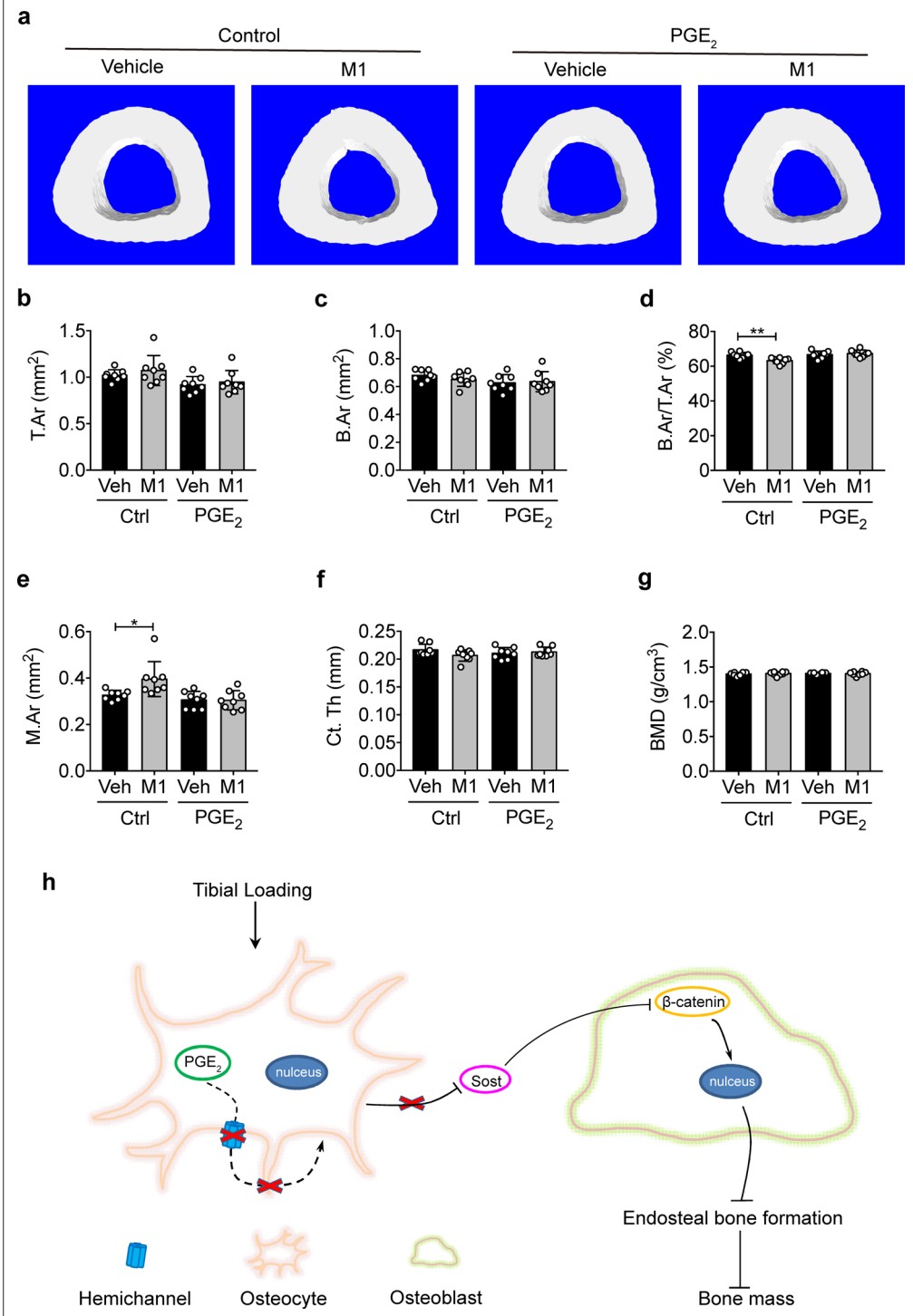

**Figure 8.** PGE$_2$ rescues the osteogenic response to mechanical loading with the impairment of Cx43 hemichannels in cortical bone. (**a**) Representative 3D models of the tibial midshaft cortical bone (50% site) in vehicle and Cx43(M1)-treated mice treated with 1 mg/kg/day PGE$_2$ or vehicle control. (**b–g**) μCT was used to assess tibial midshaft cortical bone; (**b**) total area, (**c**) bone area, (**d**) bone area fraction, (**e**) bone marrow area, (**f**) cortical thickness, and (**g**) bone mineral density. n = 8/group. Data are expressed as mean ± SD. *, p < 0.05; **, p < 0.01. Statistical analysis was performed using unpaired t-test. (**h**) Schematic diagram illustrating the mechanistic roles of osteocytic Cx43 hemichannels in mediating anabolic responses to tibial loading. Briefly, Cx43 hemichannels mediate the release of PGE$_2$ by mechanical loading, leading to suppression of SOST expression with enhanced β-catenin expression and osteogenesis on the endosteal surface. The inhibition of Cx43 hemichannels impedes

*Figure 8 continued on next page*

*Figure 8 continued*

the loading-induced PGE$_2$ secretion and anabolic function of mechanical loading on bone tissue.

The online version of this article includes the following source data and figure supplement(s) for figure 8:

**Source data 1.** Cortical micro-CT data of vehicle and Cx43(M1)-treated mice treated with 1 mg/kg/day PGE$_2$ or vehicle control.

**Figure supplement 1.** Body weights of mice of vehicle- and Cx43(M1)-treated mice treated with 1 mg/kg/day PGE$_2$ or vehicle control.

**Figure supplement 1—source data 1.** Raw data of body weight for *Figure 8—figure supplement 1a*.

**Figure supplement 2.** PGE$_2$ does not exert additional trabecular osteogenic response to mechanical loading.

**Figure supplement 2—source data 1.** Trabecular micro-CT data of vehicle and Cx43(M1)-treated mice treated with 1 mg/kg/day PGE$_2$ or vehicle control.

enhanced periosteal bone formation was further observed when hemichannels were inhibited by Cx43(M1). These results posit the role of Cx43 hemichannels in periosteal bone formation during mechanical loading. Our previous study showed that Δ130–136 mice have more periosteal bone apposition than WT and R76W mice, suggesting that periosteal osteoblasts in Δ130–136 mice are more active and sensitive than WT and R76W mice (*Xu et al., 2015*). Thus, osteocytic Cx43 hemichannels exert differential roles in controlling osteogenic osteoblastic activities on periosteal and bone resorping osteoclastic activity on endosteal surfaces, respectively. The consequence of the disruption of coordinated activities by hemichannel inhibition results in an enlarged bone marrow cavity. This is likely an adaptive response due to compromised cortical bones resulting from impaired hemichannels and consequently lower extracellular PGE$_2$. From a mechanical point of view, increased bone marrow area and cortical bone size allow the bone to respond to high stress levels (*Sharir et al., 2008*). Thus, the role of osteocytic Cx43 hemichannels in regulating the osteogenic response on endosteal surfaces is distinct from its role on periosteal surfaces.

Besides cortical bone, the anabolic response of trabecular bone to tibial loading in Δ130–136 mice was also attenuated or even reversed, as observed in trabecular number, separation and bone density. However, the compromised response of trabecular bone to mechanical loading was less evident in Cx43(M1) treated mice, except for trabecular number and trabecular separation. One possible explanation is that the accessibility and binding of the antibody to osteocytes in trabecular bone may not be as efficient as in cortical bone. Indeed, we could clearly detect Cx43(M1) on the surface of osteocytes in cortical bone. However, the binding in trabecular bone is much weaker than that in cortical bone. Since Haversian canals containing blood vessels provide supply to the osteocytes in cortical, but not in trabecular bone (*Dahl and Thompson, 2011*), it is plausible that the delivery of Cx43(M1) to the bone is mediated primarily by the Haversian canal system.

Previous studies have reported that in vitro Cx43 hemichannel opening induced by FFSS mediates the release of PGE$_2$ (*Cherian et al., 2005*), a critical factor for anabolic function of bone in response to mechanical loading (*Jee et al., 1985*; *Thorsen et al., 1996*). On the contrary, PGE$_2$ release by FFSS is inhibited by a potent hemichannel-blocking rabbit polyclonal antibody Cx43(E2) (*Siller-Jackson et al., 2008*). Moreover, we have shown that the inhibition of Cx43 hemichannels does not affect intracellular PGE$_2$ level (*Siller-Jackson et al., 2008*), suggesting the reduced PGE$_2$ biosynthesis by COX-2 since COX-2 is the enzyme subtype responding to mechanical loading. Here, we found that PGE$_2$ levels and osteocytic COX-2 expression were increased by tibial loading in WT and R76W mice, but such an increase was not detected in Δ130–136 mice. The use of hemichannel-blocking monoclonal Cx43(M1) antibody further confirmed the role of the hemichannels in the release of PGE$_2$ in bone in situ. In accordance with our observation, the reduced release of PGE$_2$ was reported in calvarial cells isolated from Cx43 cKO mice driven by the 2.3 kb *Col1α1* promoter promoter after mechanical stretching (*Grimston et al., 2006*). The increase in *Cox-2* gene expression in Cx43 cKO mice driven by the 8 kb *Dmp1* promoter is attenuated after axial tibial compression (*Grimston et al., 2012*). In addition to the regulation of PGE$_2$ release and COX-2 expression by Cx43 hemichannels, we and others have shown that PGE$_2$ can increase Cx43 expression and gap junction communication in cultured osteoblasts (*Civitelli et al., 1998*) and osteocytes (*Cheng et al., 2001a*), but has no effect in oral-derived human osteoblasts (*Adamo et al., 2001*). Additionally, increasing Cx43 expression enhances PGE$_2$-dependent β-catenin signaling activation in osteoblast cells (*Gupta et al., 2019*). Cx43 overexpression in rabbit

and human synovial fibroblast cell lines increased PTGS2 gene expression (*Gupta et al., 2014*). Moreover, increased extracellular PGE$_2$ could serve as a feedback inhibitor that activates MAPK, phosphorylates Cx43 and closes Cx43 hemichannels (*Riquelme et al., 2015*).

We showed that inhibited release of PGE$_2$ in Δ130–136 mice and in the Cx43 (M1) group was accompanied by an attenuated endosteal bone response to mechanical loading. Moreover, PGE$_2$ injection rescued the anabolic responses of cortical bone to mechanical loading impeded by Cx43(M1), including the ratio of bone area to tissue area, cortical thickness, and bone marrow area. However, the rescue of attenuated responses in cortical bone by PGE$_2$ was not shown in trabecular bone. A recent paper has also shown more beneficial osteogenic responses of combined treatment of PTH(1-34) and mechanical loading to cortical bone than trabecular bone (*Roberts et al., 2020*). One of the possibilities could be related to the higher strain levels experienced by cortical bone compared to trabecular bone.

PGE$_2$ is a skeletal anabolic factor, and its synthesis and release are highly responsive to mechanical stimulation in osteocytes (*Cherian et al., 2005*; *Jiang and Cherian, 2003*). Using the microdialysis technique, a rapid and significant increase of PGE$_2$ levels in the proximal tibial metaphysis was observed in response to dynamic mechanical loading in healthy women (*Thorsen et al., 1996*). Furthermore, intermittent PGE$_2$ treatment increases endosteal bone formation (*Jee et al., 1985*) and bone mass (*Tian et al., 2007*). Conversely, inhibition of PGE$_2$ by a COX-2 inhibitor blocks endosteal tibial bone formation induced by mechanical loading in rats (*Forwood, 1996*). Here, we demonstrate that PGE$_2$ is indeed involved in the anabolic action of hemichannels in response to mechanical loading. Interestingly, PGE$_2$ administration did not provide an additional increase in the cortical bone of mice in loaded vehicle control group. It is likely that extracellular PGE$_2$ released by osteocytes by normal exercise (mechanical loading) is sufficient to promote bone formation and additional extracellular PGE$_2$ would not further increase cortical bone mass.

Increased PGE$_2$ by mechanical stimuli is reported to bind to EP4 receptor and reduce SOST expression (*Galea et al., 2011*). SOST, a Wnt signaling antagonist (*Semënov et al., 2005*), is a key regulator of mechanotransduction in bone. SOST, secreted primarily by osteocytes, acts upon osteoblasts in a paracrine manner to inhibit bone formation (*Poole et al., 2005*) through its binding to the Wnt co-receptor Lrp5/6 (*Li et al., 2005*) and suppressing β-catenin (*Sawakami et al., 2006*). SOST gene and protein expression is suppressed by mechanical loading, and is accompanied by increased bone formation (*Moustafa et al., 2012*; *Robling et al., 2008*). We observed suppressed SOST expression in WT and R76W mice by tibial loading; however, the suppressive effect of SOST disappeared in Δ130–136 mice and the Cx43(M1)-treated mice. Correspondingly, the increased β-catenin expression and osteoblast activity observed in WT and R76W mice was abated in Δ130–136 and the Cx43(M1) mice. These results indicate that PGE$_2$ released by Cx43 hemichannels in osteocytes is a likely factor that participates in the bone anabolic response to mechanical stimuli. We previously showed that PGE$_2$ released from osteocytes via Cx43 hemichannels exerts autocrine effects via the EP2/4 receptor during mechanical stimulation (*Xia et al., 2010*). This study indicates that the increased β-catenin in osteoblasts by tibial loading is attenuated in Δ130–136 and the Cx43(M1)-treated mice. These results establish a close functional relationship between Cx43 hemichannel-released PGE$_2$ and decreased SOST, and thereby increased β-catenin expression in osteoblasts, ultimately leading to enhanced osteoblast activity and endosteal bone formation (*Figure 8h*).

There are possible limitations in this study. First, analysis of cortical bone changes at additional proximal or distal sites may provide a more comprehensive understanding of the role of Cx43 hemichannels in anabolic responses to mechanical loading, although a previous study has reported that cortical bone located at 25%, 37%, and 50% of the tibia's length had similar responses to tibial loading (*Yang et al., 2017*). Second, the monoclonal Cx43(M1) antibody blocks hemichannels not only in osteocytes, but also, possibly other cells, such as osteoblasts. However, in our study, Cx43(M1) was primarily detected in osteocytes, not in osteoblasts or other bone cells.

In summary, this study, for the first time, unveils the crucial role of osteocytic Cx43 hemichannels in mediating the anabolic function of mechanical loading on endosteal bone surfaces and trabecular bone. Cx43 hemichannels activated by mechanical stimulation release PGE$_2$ from osteocytes, which suppresses SOST expression in osteocytes, and enhances osteoblast activity and bone formation on endosteal surfaces. These results suggest that osteocytic Cx43 hemichannels could be established as a de novo new therapeutic target, and activation of these channels may potentially aid in treating bone

loss, in particular, in the elder population with the lost sensitivity to anabolic responses to mechanical stimulation (*Lanyon and Skerry, 2001*).

## Materials and methods

### Mouse models

Two transgenic models expressing dominant-negative mutants of Cx43 in osteocytes, R76W and Δ130–136, were generated as previously described (*Xu et al., 2015*). The two transgenes were driven by a 10 kb *Dmp1* promoter and expressed predominantly in osteocytes. The WT and transgenic mice in C57BL/6 J background were housed in a temperature-controlled room with a light/dark cycle of 12 hr at the University of Texas Health Science Center at San Antonio (UTHSCSA) Institutional Lab Animal Research facility under specific pathogen-free conditions. Food and water were freely available. Fifteen-week-old male WT and homozygous transgenic were sedated under isoflurane and euthanized by cervical dislocation. All animal protocols were performed following the National Institutes of Health guidelines for care and use of laboratory animals and approved by the UTHSCSA Institutional Animal Care and Use Committee (IACUC).

### Tibial mid-diaphyseal strain measurements and cyclic tibial loading

The relationship between applied compressive loading and bone tissue deformation of the left tibia was established for 15-week-old mice in vivo following a previously reported protocol (*De Souza et al., 2005*; *Lynch et al., 2010*). Briefly, a strain gauge (EA-06-015DJ-120, Vishay Measurements Group) was attached on the tibial diaphyseal medial mid-shaft of a euthanized mouse and load applied from 0 to 9.5 N at the ends of the left tibia using a loading machine (LM1, Bose). The strain gauge was connected to a bridge completion module (MR1-350-127, Vishay Measurements Group) and a conditioner/amplifier system. Mechanical load-induced strain was measured, and the compliance relationship between applied load and the resulting strain was determined for each left tibia ($R^2 > 0.99$). Compared to WT mice, a higher compressive force was required to generate comparable periosteal strain in Δ130–136 mice (*Figure 1—figure supplement 1c*).

The cyclic axial compressive load was applied to the left tibia of each mouse using a custom loading device based on previous studies (*De Souza et al., 2005*; *Lynch et al., 2010*). Briefly, the left tibia of anesthetized mice was positioned into a custom-made apparatus (*Figure 1—figure supplement 1a*). The upper padded cup containing the knee was connected to the loading device (7528–10, Masterflex L/S, Vernon Hills, IL, USA), and the lower cup held the heel. The left tibia was held in place by a 0.5 N continuous static preload, loaded for 600 cycles (5 min) at 2 Hz frequency, with a sinusoidal waveform (*Figure 1—figure supplement 1b*). Compressive load was performed 5 days/week for 2 weeks to determine bone structural and anabolic response, or 5 consecutive days to assess $PGE_2$ level. Based on the load-strain relationship, peak force was selected to generate peak periosteal strains of 1200 $\mu\varepsilon$ at the cortical midshaft for WT, R76W, and Δ130–136 mice, respectively. This strain level has been previously shown to elicit an anabolic response at this region (*Melville et al., 2015*). The right tibia was used as a contralateral, non-loaded control.

### M1 antibody generation and treatment

A monoclonal Cx43(M1) antibody targeting the second extracellular loop domains of Cx43 was originally generated by Abmart (Tulsa, OK, USA) and described previously (*Zhang et al., 2021*). Briefly, mice were immunized with a Cx43 extracellular domain peptide, and after functional characterization of the hybridoma clones, genes that encode the antibody heavy and light chain variable region were cloned from the mouse hybridoma cell line HC1 by reverse transcription quantitative PCR (RT-qPCR), using a combination of a group of cloning PCR primers. The heavy and light chain constructs were co-transfected into human embryonic kidney freestyle 293 (HEK293F) cells, supernatants were harvested, and antibodies were purified by affinity chromatography using protein A resin.

The day before tibial loading, randomly allocated WT mice based on the body weight were intraperitoneally (IP) injected with 25 mg/kg Cx43(M1) or vehicle (phosphate-buffered saline (PBS), pH 7.4). A second dose was administered the day before the start of loading in the 2nd week. The dosage of antibody was based on our data with Cx43(M1) antibody (**Figure S5**) and a previous study using an anti-sclerostin (SOST) antibody (*Spatz et al., 2013*).

## Cell lines

The osteocyte-like MLO-Y4 cell line was originally obtained from Dr. Lynda Bonewald at Indiana University. We maintain this cell line strictly with proper culture condition, collagen coating and cell density (70–80% confluence for cell passage). We routinely conduct quality control of this cell line including testing for mycoplasma contamination and examining cell morphology with dendritic processes and maker protein expression, low amount of alkaline phosphates and high amount of osteocalcin, Cx43 and CD44. We are ensured to use the cell with these characteristics for all the experiments.

## In vitro dye uptake and Gap junction coupling assays

MLO-Y4 cells were cultured in a-modified essential medium (a-MEM) with 2.5% fetal bovine serum (FBS) and 2.5% calf serum (CS) in a 5% $CO_2$ incubator at 37 °C. MLO-Y4 cells were grown at a low initial cell density on glass slides coated with type I collagen (rat tail collagen type I, Corning, Bedford, MA, USA, 0.15 mg/ml) to ensure that most of the cells were not physically in contact. The cells were preincubated with Cx43(E2) or Cx43(M1) (2 µg/ml) for 30 min and then subjected to FFSS at four dynes/$cm^2$ for 15 min in the presence of 25 mM ethidium bromide (EtBr) in the recording media ($HCO_3^-$-free a-MEM medium buffered with 10 mM HEPES). These cells were then fixed with 1% paraformaldehyde (PFA) for 10 min. The intensity of EtBr fluorescence in cells was measured and quantified by NIH Image J software (NIH, USA). Primary osteocytes were microinjected using an Eppendorf micromanipulator InjectManNI two and Femtojet (Eppendorf) at 37 °C with 10 mM Oregon green 488 BAPTA-AM (Mr: 1751 Da) as a cell tracker probe, and calcein red-orange AM (Mr: 789 Da) as a probe for detecting gap junction coupling. Images were captured using an inverted microscope equipped with a Lambda DG4 device (Sutter Instrument Co, Novato, CA, USA), a mercury arc lamp illumination, and a Nikon Eclipse microscope (Nikon, Tokyo, Japan) using a rhodamine filter. Loaded cells (dye donor) were 'parachuted' over acceptor cells. The cells (acceptors and donors) were pre-incubated for 20 min with Cx43 antibodies before the parachuting assay. Donor cells were then incubated with acceptor cells for 90 min, the time duration sufficient to detect dye transfer.

## In vivo dye uptake assay

We developed an approach to assess hemichannel activity in osteocytes in the bone in vivo (*Riquelme et al., 2021*). Briefly, 20 mg/ml Evans blue (EB) dye dissolved in sterile saline solution (previously used to study hemichannel activity in muscle cells in vivo *Cea et al., 2013*) was injected into the mouse tail vein. For the vehicle or Cx43(M1) treated group, mice were IP injected with mouse IgG or Cx43 (M1) (25 mg/kg) 4 hr before dye injection. After the dye injection, mice were kept in cages for 20 min, and the left tibias were then loaded for 10 min. Mice were sacrificed and perfused with PBS and 4% PFA 40 min after tibial loading. Tibias were isolated and fixed in 4% PFA for 2 days, decalcified in 10% ethylenediaminetetraacetic acid (EDTA) for 3 weeks, and then 12-µm-thick frozen sections were prepared. The cell nuclei were stained with 4',6-diami-dino-2-phenylindole (DAPI). Images were captured using an optical microscope (BZ-X710, KEYENCE, Itasca, IL, USA) and EB fluorescence intensity in osteocytes was quantified by NIH Image J software (NIH, USA).

## PGE₂ measurement and treatment

The level of $PGE_2$ in the tibia bone was determined according to the manufacturer's protocol ($PGE_2$ ELISA kit, #514010, Cayman Chemical, Ann Arbor, MI, USA). Briefly, 4 hr after the final round of five-day tibial loading, bone marrow-flushed tibias were isolated free of soft tissues, and bone shafts were prepared by removing proximal and distal ends of the bone. Bone tissue was homogenized in liquid nitrogen with a frozen mortar and pestle. The concentration of $PGE_2$ was normalized by total protein concentration using a BCA assay (#23225, Thermo Scientific, Rockford, IL, USA).

$PGE_2$ powder (#2296, Tocris Bioscience, Bristol, UK) was dissolved in 10% ethanol and stock prepared at the concentration of 0.15 mg/ml. During the two-week cyclic tibial loading (5 days/week), wild-type mice randomly allocated based on the body weight were injected with 1 mg/kg/day of $PGE_2$ solution or 6.7 µl/kg/day 10% ethanol (vehicle) at a fixed time everyday.

## Micro-computed tomography

Tibias were dissected and frozen in saline-soaked gauze at –20 °C until scanning. Samples in PBS were imaged using a high-resolution micro-computed tomography (µCT) scanner (1172, SkyScan,

Brüker microCT, Kontich, Belgium) with the following settings: 59 Kvp, 167 µA beam intensity, 0.5 mm aluminum filter, 800ms exposure, 1024 × 1024 pixel matrix, and a 10 µm isotropic voxel dimension. The background noise was removed from the images by eliminating disconnected objects smaller than four pixels in size. Two bone volumes of interest (VOI) were selected in the metaphyseal and midshaft regions. The analyses were conducted excluding the fibula. In the proximal tibial metaphysis, the trabecular bone VOI was positioned 0.44 mm distal to the proximal growth plate and extended 0.65 mm in the distal direction, excluding the primary spongiosa. Grayscale values of 80–256 were set as the threshold for trabecular bone. For the cortical region, a 0.3 mm distance was centered at 50% tibial length (proximal to distal). Automated contouring was used to select the cortex. A threshold of 106–256 was applied to all of the cortical slices for analysis. The structural morphometric properties of cortical and trabecular regions were analyzed using the CT Analyser software (CTAn 1.18.8.0, Bruker Skyscan).

## Mechanical testing

Three-point bending tests were performed after µCT scanning. The tibia was thawed to room temperature before testing. Any remaining muscles and the fibula were carefully removed. The tibia was subjected to a three-point bending test along the medial-lateral direction in a micromechanical testing system (Mach-1 V500CST, Biomomentum, Laval, Canada). The span distance for the three-point bending test was 8 mm, and the loading pin was placed at the midpoint of the span. The test was performed in a displacement control mode at a constant rate of 0.01 mm/s, and the data was collected at a 200 Hz sampling rate for all measurements. The accurate cross-sectional areas were determined from µCT and used to calculate mechanical properties (*Jepsen et al., 2015*).

## Histomorphometry, immunohistochemistry, and dynamic bone histomorphometry

Tibias were collected and fixed in 4% PFA for 2 days and decalcified using 10% EDTA (pH 7.5) for 21 days. These tibial samples were embedded in paraffin, and 5-mm-thick sections were collected and mounted onto glass slides. For static bone histomorphometry, tartrate resistant acid phosphatase (TRAP) and toluidine blue staining was used to determine the osteoclast activity (*Xu et al., 2015*) and osteoblast numbers, respectively. For immunohistochemistry, paraffin sections were rehydrated and antigen site was retrived with 10 mM citrate buffer (pH 6.0) at 60 °C for 2 hr (for sclerostin or β-catenin) or trypsin buffer (pH 7.8) at 37 °C for 30 min (for COX-2), and then probed with an anti-sclerostin (AF1589, 1:400, R&D systems, Minneapolis, MN, USA), anti-COX-2 (12375–1-AP, 1:200, Proteintech, Rosemont, IL, USA) or an anti-β-catenin antibody (ab16051, 1:200, Abcam, Waltham, MA, USA) overnight at 4 °C. The sections were probed with a biotin-labeled secondary antibody and ABC Reagent (VECTASTAIN, Burlingame, CA, USA). Staining was visualized with DAB Chromogen (SK-4100, Vector Laboratories, Burlingame, CA, USA). Hematoxylin was used as a counterstain. Images were captured using an optical microscope (BZ-X710, KEYENCE). Osteoclast surface (Oc.S), osteoclast number (N.Oc), osteoblast number (N.Ob), and bone surface (BS) on the tibial midshaft cortical bone along the endosteal surface were counted and positive cells quantified using ImageJ software (NIH, USA).

Mice were IP injected with calcein (C0875, Sigma-Aldrich, St. Louis, MO, USA) at 20 mg/kg of body weight 1 day before tibial loading, and followed by alizarin red injection (A5533, Sigma-Aldrich, St. Louis, MO, USA) at 30 mg/kg of body weight 3 days before euthanization. Tibias were dissected, fixed in 70% ethanol, and then embedded in methylmethacrylate for 10-µm-thick longitudinal plastic sections. Two-color fluorescent images were obtained using a fluorescence microscope (BZ-X710, KEYENCE). Single label was defined as any bone surface with green, red, or yellow (no separation between green and red). The distance between green and red fluorescence signals was measured along the bone surface (*Grimston et al., 2012*). The following parameters were quantified at tibial midshaft using NIH ImageJ software (NIH, USA): total perimeter (BS); single label perimeter (sLS); double label perimeter (dLS), and double-label area (dL.Ar). The following values were then calculated: mineralizing surface [MS/BS = (sLS/2+ dLS)/BS], mineral apposition rate [MAR = dL.Ar/dLS/12], and bone formation rate (BFR/BS = MAR × MS/BS).

## RNA extraction and RT-qPCR

The long tibial bone was isolated free of soft tissues, and bone marrow was removed by flushing with RNase-free PBS after two weeks tibial loading. The bone shaft was prepared by removing the proximal and distal ends of the bone and pulverizing it using a frozen mortar and pestling in liquid nitrogen. Total RNA was isolated by using TRIzol (Molecular Research Center, Cincinnati, OH, USA) according to the manufacturer's protocol. cDNA was synthesized by a high-capacity cDNA reverse transcription kit (#4388950, Applied Biosystems, Carlsbad, CA, USA). mRNA level was analyzed by real-time RT-qPCR using an ABI 7900 PCR device (Applied Biosystems, Bedford, MA, USA) and SYBR Green (#1725124, Bio-Rad Laboratories, Hercules, CA, USA) with a two-step protocol (94 °C for 10 s, and 65 °C for 30 s for 40 cycles). Relative mRNA expression levels were normalized to GAPDH by using the ΔCt method (*Livak and Schmittgen, 2001*). The primers for *Sost, Ptgs2, Runx2, Bgalp, Ctnnb1,* and *Dmp1* are provided in *Supplementary file 1*.

## Statistical analysis

Data collection and analysis were conducted in a blind manner. Statistical analysis was performed using IBM SPSS Statistics 24 (SPSS Inc, Chicago, IL, USA) and graphed with GraphPad Prism 7 (GraphPad Software; La Jolla, CA, USA). For in vitro studies, each experiment had three technical replicates and was repeated at least three times. Normal distribution of the data was evaluated by the Shapiro-Wilk test, and homogeneity of variance was assessed by the Levene test. The paired t-test was used for comparisons of the loaded and contralateral tibias within the same group. One-way ANOVA with Tukey test was used for multiple group comparisons. Student unpaired t-test was used to compare between vehicle and Cx43 (M1)-treated groups. All data are presented as means ± SD. $p < 0.05$ was considered significant.

## Acknowledgements

We thank Dr. Songqing Lu at UTHSCSA for assistance in statistical analysis, Dr. Lynda Bonewald at Inidiana University for generously providing MLO-Y4 cell line, and Dr. Eduardo Cardenas and Dr. Francisca Acosta at UTHSCSA for critical reading and editing of the paper. Funding: This work was supported by the National Institutes of Health (NIH) Grants: 5RO1 AR072020 (to JXJ), and Welch Foundation grant: AQ-1507 (to JXJ). We also thank the support of the UTHSCSA CMMI and the UTHSCSA Optical Imaging Facility supported by the Cancer Therapy and Research Center through NIH–National Cancer Institute P30 award CA054174 and Texas State funds.

## Additional information

### Funding

| Funder | Grant reference number | Author |
| --- | --- | --- |
| National Institutes of Health | AR072020 | Jean Jiang |
| Welch Foundation | AQ-1507 | Jean Jiang |

The funders had no role in study design, data collection and interpretation, or the decision to submit the work for publication.

### Author contributions

Dezhi Zhao, Conceptualization, Data curation, Formal analysis, Investigation, Methodology, Writing – original draft, Writing – review and editing; Manuel A Riquelme, Conceptualization, Data curation, Investigation, Methodology, Validation, Writing – review and editing; Teja Guda, Chao Tu, Huiyun Xu, Data curation, Investigation, Methodology, Writing – review and editing; Sumin Gu, Conceptualization, Data curation, Investigation, Methodology, Writing – review and editing; Jean X Jiang, Conceptualization, Formal analysis, Funding acquisition, Investigation, Project administration, Resources, Supervision, Writing – original draft, Writing – review and editing

## Author ORCIDs
Dezhi Zhao http://orcid.org/0000-0001-8249-8743
Manuel A Riquelme http://orcid.org/0000-0002-1915-0434
Jean X Jiang http://orcid.org/0000-0002-2185-5716

## Ethics

This study was performed in strict accordance with the recommendations in the Guide for the Care and Use of Laboratory Animals of the National Institutes of Health. All of the animals were handled according to approved institutional animal care and use committee (IACUC) protocols (#20060124AR) of the University of Texas Health Science Center at San Antonio. All surgery was performed under sodium pentobarbital anesthesia, and every effort was made to minimize suffering.

## Decision letter and Author response
Decision letter https://doi.org/10.7554/eLife.74365.sa1
Author response https://doi.org/10.7554/eLife.74365.sa2

## Additional files

### Supplementary files
• Supplementary file 1. Sequences of the primers used for each gene used in this article.
• Transparent reporting form

### Data availability
Source data has been provided.

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
