## [Editor Report]

This study documents a key role for osteocyte-derived Cx43 hemichannels in the anabolic response of bone to mechanical stimulation, using transgenic mouse models and a Cx43 (M1) antibody. The studies shed new light on the molecular mechanisms that transduce mechanical loading into new bone formation, and identify Cx43 as an actionable target.

---

## [Decision Letter]

**Decision letter after peer review:**

[Editors’ note: the authors submitted for reconsideration following the decision after peer review. What follows is the decision letter after the first round of review.]

Thank you for submitting the paper "Connexin Hemichannels with Prostaglandin Release in Anabolic Function of Bone to Mechanical Loading" for consideration by *eLife*. Your article has been reviewed by 2 peer reviewers, and the evaluation has been overseen by a Reviewing Editor and a Senior Editor. The reviewers have opted to remain anonymous.

We are sorry to say that, after consultation with the reviewers, we have decided that the current manuscript will not be considered further for publication by *eLife*. However, we would be open to considering a new manuscript that addresses all of the review concerns, but without any guarantee on the outcome of the review process.

Comments to the Authors:

Overall, the study was found to be limited in novelty and impact. The use of Evans Blue dye entry to measure hemichannel activity in osteocytes may not be appropriate because of the observed membrane tears in osteocytes caused by mechanical loads. Furthermore, the very small effect size seen in many of the skeletal parameters measured in response to mechanical loading coupled with the in correct statistical analyses used raises concerns regarding the validity of the conclusions drawn.

*Reviewer #1:*

I'm not sure why the authors are not seeing Evans Blue dye entry into the osteocytes of loaded bone from D130-136 transgenic mice. The Augusta, GA very nicely (and it has since been repeated) that osteocyte membrane disruptions occur with much milder loading (e.g., treadmill running) and allow in EB dye. These membrane tears have nothing to do with channel or hemichannel activity. So it is very hard to understand why the D130-136 mice would be spared from membrane tears that should allow copious amounts of EB into the cells. Do certain mutations in connexin prevent membrane tears?

If the R76W mutation enhances hemichannel function, and the conclusions of the paper are correct that the hemichannels are controlling the response to loading, then why were the R76W mutants not more responsive than WT to mechanical loading?

Figure 3: How is it justified to say that the D130-D136 mice had increased bone formation response to loading on the periosteum when the relative change between loaded and nonloaded look to be about the same in all three genotypes? Are the authors not adjusting for the higher or lower control leg bone formation measurements?

*Reviewer #2:*

This study examines the effects of mechanical loading on the bones of two transgenic mouse models of connexin 43 overexpression, one mutant which impairs both gap junction intercellular communication (GJIC) and hemichannel activity (∆130-136) and another that supports only enhanced hemichannel activity but not GJIC (R76W). The authors conclude that hemichannels but not GJIC facilitate the effects of mechanical loading on bone via the secretion of PGE2 through the hemichannels.

While provocative, the data fall short of being convincing of the interpretation.

A major concern is the statistical approaches used to evaluate data. The conclusions obligate that each group of animals (WT, R76W and ∆130-136 mice with or without loading) be compared to each other to determine differences in their ability to mount a response of bone to a mechanical load. The correct statistical test is a two way ANOVA when there are multiple variables (genotype and load). However, multiple t-tests are used to support major conclusions. Since primary data was supplied by the authors in the supplement, we checked this using statistical software. Many of the statistical analyses do not hold up when run through the appropriate statistical test. Thus, the primary findings reported are not supported.

Two additional significant weaknesses affect the potential quality and impact of this study.

1. No convincing evidence is presented that the phenotype was rescued by PGE2. In Figure 8 and the corresponding supplement, vehicle treated and PGE2 treated unloaded controls are not shown and are critical to the appropriate interpretation of the experiment. Meaningful bone parameters including bone area and cortical thickness are not affected by the PGE2. Trabecular bone was completely unaffected by PGE2 or even the M1 antibody. Also, a one-way ANOVA is the incorrect measure with which to assess these changes. There are many variables in these mice: treatment with or without M1 antibody, loading or unloading (although not included) and treatment with or without PGE2. These are not accounted for with the statistical models used to assess the data.

2. No convincing evidence that PGE2 secretion through connexin 43 hemichannels is shown. Instead, Figure 4C shows that a protein (COX2) responsible for producing PGE2 is reduced in the cells that produce PGE2 in the D130-136 mice. Several papers have shown that connexin 43 regulates ptgs2 and could affect PGE2 abundance independent of the ability to pass through connexin 43 hemichannels and others show that PGE2 also regulates connexin 43 abundance and gap junctional communication.

As mentioned above, most of the statistical tests used to analyze the data are incorrectly performed throughout the manuscript. This prevents any meaningful conclusions from large portions of the study.

The novelty and impact of these studies is limited. As indicated by the authors, the concept that connexin 43 hemichannels are involved in PGE2 release by osteocytes was reported {greater than or equal to}15 years ago (Cherian et al., 2005; Jiang and Cherian, 2003) and has become accepted in the bone literature: "Connexin (Cx)-forming gap junctions and hemichannels permit small molecules ({less than or equal to}1 kDa) to pass through the cellular membrane, such as prostaglandin E2 (PGE2) and ATP (Loiselle et al., 2012)." These same transgenic mouse models have been reported for unloading (Zhao et al., Front Physiol, 2020), estrogen deficiency (Ma et al., Bone Res 2019), fracture healing (Chen et al., J Cell Physiol 2019), breast cancer metastasis (Zhou et al., Oncogene 2016), and muscle-bone crosstalk via PGE2 ( Li et al., Cells 2021), and in normal bone acquisition (Xu et al., J Bone Miner res 2015). Adding loading is important to the bone field but may not be of broad interest. The authors should justify/clarify what aspect of this study will be broadly impactful or what knowledge gap is filled.

Controls demonstrating that the R76W protein is expressed are required. Evidence that these mice exhibit a loss of gap junctions in vivo is needed. This control is important, since there is no distinguishable phenotype between R76W and wild type mice despite a loss of connexin 43 GJIC and gain of hemichannel activity. In a similar GJIC deficient mutant with a gain of hemichannel activity there is a severe bone phenotype (Watkins et al., Mol Biol Cell 2011; Dobrowolski Hum Mol Genet 2008). This is in stark contrast to the R76W mutant used here.

Controls showing that the long term treatment with M1 antibodies blocks hemichannels but not gap junctions in vivo is required. It is expected that two weeks of antibody treatment, which binds to the extracellular domains of connexin 43, would likely result in steric hindrance to the docking of connexin hemichannels to form gap junctions. The phenotype of these chronically treated mice beings to resemble connexin 43 gene deletion (Figure 6H increased periosteal bone formation rate) – a key feature of loss of GJIC (D130-136 mice or bone connexin 43 knockout mice). Increased endosteal osteoclast activity was also found in the Cx43(M1) group (Figure 7K, N, O) also consistent with complete Cx43 (gap junction and hemichannel) loss of function.

Figure 4 F, I, J, K and Figure 7D, G, J should be shown with values for loaded and unloaded gene expression rather than a ratio. Then a two-way ANOVA (loading and genotype) should be used to assess the effects. It is statistically unsound as shown.

Figure 1 FS2 is too dark to assess. Please supply a higher contrast image.

Figure 5—figure supplement 1A quantification and statistical analysis are required.

In the section titled: "PGE2 rescues impeded osteogenic responses to mechanical loading by impaired Cx43 hemichannels." It is concluded that "Together, these results demonstrate that administration of PGE2 significantly rescues impeded anabolic responses of cortical bone to mechanical loading as a result of Cx43 hemichannel inhibition." This is not supported by the data.

---

## [Author Response]

[Editors’ note: The authors appealed the original decision. What follows is the authors’ response to the first round of review.]

Overall, the study was found to be limited in novelty and impact.

We respectfully disagree with this assessment. Indeed hemichannels as mechanosensitive channels with PGE_2_ release in osteocytes have been well studied in osteocyte-like MLO-Y4 cells including our previously published papers. However, the study reported in this paper, for the first time, demonstrated the in vivo role of Cx43 hemichannels in osteocytes and PGE_2_ release in mediating anabolic function in the bone. The outcomes of this study will help establish hemichannels as a potential new drug target for treating bone loss and osteoporosis.

The use of Evans Blue dye entry to measure hemichannel activity in osteocytes may not be appropriate because of the observed membrane tears in osteocytes caused by mechanical loads.

Hemichannels are selective channels that only allow molecular weight less than 1 kDa molecules to pass through, while membrane tear has no size limit for passing molecules. Therefore, two types of fluorescence dyes are typically used to study Cx hemichannels, one is less than 1 kDa and the other is around 10 kDa to distinguish hemichannels from other forms of dye uptake like membrane tear. The uptake detected by 10 kDa is very low. In addition to transgenic mouse model with impaired hemichannels, we also used a specific Cx43 hemichannel blocking monoclonal antibody M1 and it can significantly block uptake of small sized dye molecule. Other previous studies including us also demonstrate the inhibition of hemichannel opening induced by mechanical loading with other inhibitors including chemicals and Cx mimic peptides both in vitro and in vivo. However, currently there is no any specific inhibitor that can inhibit dye uptake associated with membrane tears.

Furthermore, the very small effect size seen in many of the skeletal parameters measured in response to mechanical loading coupled with the in correct statistical analyses used raises concerns regarding the validity of the conclusions drawn.

Unlike gene knockout mouse models, anabolic effects by mechanical loading for two weeks, a standard protocol, normally does not cause very dramatic bone responses, but does show the consistent anabolic responses. The bone anabolic responses shown in our study is consistent with other published studies (Bass et al., 2002; Zhang et al., 2011).

In the revision, we have reviewed all of our statistical analysis with a biostatistician expert. We have removed one-way ANOVA and used student T-test for paired comparisons. The main comparison in this paper is for unloaded and loaded group within the same genotypes and M1 or PGE2 for control and treated groups. The conclusion from these analyses is supported by corrected statistical analyses.

Reviewer #1:I'm not sure why the authors are not seeing Evans Blue dye entry into the osteocytes of loaded bone from D130-136 transgenic mice. The Augusta, GA very nicely (and it has since been repeated) that osteocyte membrane disruptions occur with much milder loading (e.g., treadmill running) and allow in EB dye. These membrane tears have nothing to do with channel or hemichannel activity. So it is very hard to understand why the D130-136 mice would be spared from membrane tears that should allow copious amounts of EB into the cells. Do certain mutations in connexin prevent membrane tears?If the R76W mutation enhances hemichannel function, and the conclusions of the paper are correct that the hemichannels are controlling the response to loading, then why were the R76W mutants not more responsive than WT to mechanical loading?

There are uncertainties regarding how common the phenotypes of membrane tears occurred in the cells in the referred study since no specific inhibitor and underlying mechanism are currently available. Hemichannels have been investigated for years and they are selective channels that allow molecular weight less than 1 kDa to pass through. In addition to specific hemichannel-blocking antibodies, and other hemichannel blockers, such as chemicals, such as carboxlone and connexin extracellular mimetic peptides showed the inhibition of smaller dye (i.e. EB, EtBr, Lucifer yellow) uptake, but not bigger dyes (i.e. rhodamine dextran ~10 kDa) both in vitro and in vivo.

It is true that R76W mutation has enhanced hemichannel function and some anabolic bone responses as compared to WT are indeed enhanced in R76W mice including bone volume fraction, trabecular thickness and BMD, although not as dramatic as expected. It is possible that certain threshold of hemichannel activity is required for the anabolic function in response to mechanical loading and excess hemichannel activity can be attenuated by a feedback inhibition mechanism. Our earlier study showed that prolonged activity of osteocytic Cx43 hemichannels increases extracellular PGE2 level and excess extracellular PGE2 acting in an autocrine manner activates EP2/4 receptors, leading to MAPK activation. MAPK directly phosphorylates Cx43 and closes hemichannels (Riquelme et al., 2015). This mechanism could similarly regulate the activity of R76W, resulting in comparable anabolic responses to mechanical loading as WT. We have included the above in the Discussion.

Figure 3: How is it justified to say that the D130-D136 mice had increased bone formation response to loading on the periosteum when the relative change between loaded and nonloaded look to be about the same in all three genotypes? Are the authors not adjusting for the higher or lower control leg bone formation measurements?

In WT and R76W mice, the bone formation in both periosteal and endosteal surface were increased by tibial loading and this increase is correlated with the increase of bone area fractions and cortical thickness. In D130-136 mice, only bone formation on the periosteal surface increased, but not on the endosteal surface. In addition, we observed the increased osteoclast number in endosteal surface. The net effect in D130-136 is a decreased bone area fraction and cortical thickness. The data presented in this study include the higher or lower control leg formation measurements. We have revised the text in the Discussion to make it clear.

Reviewer #2:This study examines the effects of mechanical loading on the bones of two transgenic mouse models of connexin 43 overexpression, one mutant which impairs both gap junction intercellular communication (GJIC) and hemichannel activity (∆130-136) and another that supports only enhanced hemichannel activity but not GJIC (R76W). The authors conclude that hemichannels but not GJIC facilitate the effects of mechanical loading on bone via the secretion of PGE2 through the hemichannels.While provocative, the data fall short of being convincing of the interpretation.A major concern is the statistical approaches used to evaluate data. The conclusions obligate that each group of animals (WT, R76W and ∆130-136 mice with or without loading) be compared to each other to determine differences in their ability to mount a response of bone to a mechanical load. The correct statistical test is a two way ANOVA when there are multiple variables (genotype and load). However, multiple t-tests are used to support major conclusions. Since primary data was supplied by the authors in the supplement, we checked this using statistical software. Many of the statistical analyses do not hold up when run through the appropriate statistical test. Thus, the primary findings reported are not supported.

By working closely with a biostatistician expert, in the revision, we have thoroughly reanalyzed the data with statistical analyses. To determine the mechanical responses, the major analysis should be the paired comparison within each genotype group, WT, R76W and D130-136. Therefore, paired student T-test is an appropriate statistical approach. We agree that one-way ANOVA is improper to compare multiple variables and comparison with multiple variable (genotype and load) would provide irrelevant information regarding the treatment responses. In this study, we focus on the comparison between loaded and contralateral, unloaded tibias within each genotype using paired student T-test.

Two additional significant weaknesses affect the potential quality and impact of this study.1. No convincing evidence is presented that the phenotype was rescued by PGE2. In Figure 8 and the corresponding supplement, vehicle treated and PGE2 treated unloaded controls are not shown and are critical to the appropriate interpretation of the experiment. Meaningful bone parameters including bone area and cortical thickness are not affected by the PGE2. Trabecular bone was completely unaffected by PGE2 or even the M1 antibody. Also, a one-way ANOVA is the incorrect measure with which to assess these changes. There are many variables in these mice: treatment with or without M1 antibody, loading or unloading (although not included) and treatment with or without PGE2. These are not accounted for with the statistical models used to assess the data.

The significant reduction of bone area fraction, a key parameter by M1 was ablated with PGE2 treatment as well bone marrow area and cortical thickness. As the reviewer pointed out, the rescue by PGE2 in cortical bone was not shown in trabecular bone. We are not certain for the difference between cortical and trabecular bones. A recent paper has also shown more beneficial osteogenic responses of combined treatment of PTH(1-34) and mechanical loading to cortical bone than trabecular bone (Roberts et al., 2020). As discussed in this paper, one of the possibilities could be related to the higher strain levels experienced by cortical bone compared to trabecular bone. We have included the above in the Discussion. We have reanalyzed the data and comparison. The major comparison should be paired student-T test by comparing vehicle and M1 treated within each group, Control and PGE2.

2. No convincing evidence that PGE2 secretion through connexin 43 hemichannels is shown. Instead, Figure 4C shows that a protein (COX2) responsible for producing PGE2 is reduced in the cells that produce PGE2 in the D130-136 mice. Several papers have shown that connexin 43 regulates ptgs2 and could affect PGE2 abundance independent of the ability to pass through connexin 43 hemichannels and others show that PGE2 also regulates connexin 43 abundance and gap junctional communication.

Our earlier study has showed that Cx43 hemichannels in osteocytes serve as a direct portal for the release of PGE2 (Cherian et al., 2005). In this study, we showed that increased PGE2 in tibia bone by mechanical loading was totally attenuated in D130-136 mice (Figure 4A) and M1 treated mice (Figure 7A). Moreover, we have previous reported that the inhibition of Cx43 hemichannels does not affect intracellular PGE2 level (Siller-Jackson et al., 2008), suggesting the reduced PGE2 biosynthesis by COX2 since COX2 is the enzyme subtype responding to mechanical loading. Indeed, in this study, we showed the attenuated upregulation of COX2 expression and reduction of PGE2 level in D130-136 mice as well as in M1 treated mice. We did not find any previous papers raised by the reviewer regarding “connexin 43 regulates ptgs2 and could affect PGE2 abundance independent of the ability to pass through connexin 43 hemichannels”. We and others have shown that PGE2 can increase Cx43 expression and gap junction communication in cultured osteoblasts (Civitelli et al., 1998) and osteocytes (Cheng et al., 2001), but has no effect in oral-derived human osteoblasts (Adamo et al., 2001). Additionally, increasing Cx43 expression enhances PGE2-dependent β-catenin signaling activation in osteoblast cells (Gupta et al., 2019). Cx43 overexpression in rabbit and human synovial fibroblast cell lines increased PTGS2 gene expression (Gupta et al., 2014). Moreover, increased extracellular PGE2 could serve as a feedback inhibitor that activates MAPK, phosphorylates Cx43 and closes Cx43 hemichannels (Riquelme et al., 2015). The outcomes of this study will help establish hemichannels as a potential de novo drug target for treating bone loss and osteoporosis. We have included the above in the Discussion.

As mentioned above, most of the statistical tests used to analyze the data are incorrectly performed throughout the manuscript. This prevents any meaningful conclusions from large portions of the study.

By working closely with a biostatistician expert, in the revision, we have thoroughly reanalyzed the data with statistical analyses as described in detail the responses above. After reanalysis, the primary findings and conclusion reported in the paper are fully supported.

The novelty and impact of these studies is limited. As indicated by the authors, the concept that connexin 43 hemichannels are involved in PGE2 release by osteocytes was reported {greater than or equal to}15 years ago (Cherian et al., 2005; Jiang and Cherian, 2003) and has become accepted in the bone literature: "Connexin (Cx)-forming gap junctions and hemichannels permit small molecules ({less than or equal to}1 kDa) to pass through the cellular membrane, such as prostaglandin E2 (PGE2) and ATP (Loiselle et al., 2012)." These same transgenic mouse models have been reported for unloading (Zhao et al., Front Physiol, 2020), estrogen deficiency (Ma et al., Bone Res 2019), fracture healing (Chen et al., J Cell Physiol 2019), breast cancer metastasis (Zhou et al., Oncogene 2016), and muscle-bone crosstalk via PGE2 ( Li et al., Cells 2021), and in normal bone acquisition (Xu et al., J Bone Miner res 2015). Adding loading is important to the bone field but may not be of broad interest. The authors should justify/clarify what aspect of this study will be broadly impactful or what knowledge gap is filled.

As reviewer pointed out, PGE2 release via Cx43 hemichannels in cultured MLO-Y4 cells in response to fluid flow shear stress and transgenic mouse models have been reported in our previous publications. However, this study, for the first time, demonstrated the in vivo physiological role of Cx43 hemichannels in osteocytes with PGE2 release in mediating anabolic function in the bone using both transgenic mouse models as well as hemichannel-blocking monoclonal M1 antibody. Additionally, PGE2 administration rescues the cortical defects caused by impaired Cx43 hemichannels. This is the first study unveiling the physiological role of osteocytic hemichannels in mediating the anabolic function of mechanical loading. We have revised the text and included the above discussion to justify the broad impact of this study.

Controls demonstrating that the R76W protein is expressed are required. Evidence that these mice exhibit a loss of gap junctions in vivo is needed. This control is important, since there is no distinguishable phenotype between R76W and wild type mice despite a loss of connexin 43 GJIC and gain of hemichannel activity. In a similar GJIC deficient mutant with a gain of hemichannel activity there is a severe bone phenotype (Watkins et al., Mol Biol Cell 2011; Dobrowolski Hum Mol Genet 2008). This is in stark contrast to the R76W mutant used here.

In our previous study we have shown the expression of R76W protein (Xu et al., 2015). We have also shown the impaired gap junction channels in the isolated primary osteocytes from R76W and D130-136 mice using microinjection of fluorescence dye as compared to functional gap junction channels in wild type (Xu et al., 2015). It would be ideal if we could determine gap junctions in situ, but so far, it is not technically feasible albeit the research of gap junction channels for 5 decades. The reviewer raised a good point regarding R76W mice vs reported G138R mice. Cx43 G138R is a dominant negative mutant, which impairs gap junctions with leaky hemichannels as reported in the referred references. In contrast to G138R mutant that exhibits leaky hemichannels without any stimulation, hemichannels formed by R76W is not open without mechanical loading (Xu et al., 2015). Therefore, unlike G138R mutant, compared with WT, R76W does not cause any major bone phenotypes. Under mechanical loading in vivo, further increase of hemichannel opening in R76W has stronger anabolic effects on the bone including bone volume fraction, trabecular thickness and BMD. We have included the above in the Discussion.

Controls showing that the long term treatment with M1 antibodies blocks hemichannels but not gap junctions in vivo is required. It is expected that two weeks of antibody treatment, which binds to the extracellular domains of connexin 43, would likely result in steric hindrance to the docking of connexin hemichannels to form gap junctions. The phenotype of these chronically treated mice beings to resemble connexin 43 gene deletion (Figure 6H increased periosteal bone formation rate) – a key feature of loss of GJIC (D130-136 mice or bone connexin 43 knockout mice). Increased endosteal osteoclast activity was also found in the Cx43(M1) group (Figure 7K, N, O) also consistent with complete Cx43 (gap junction and hemichannel) loss of function.

As stated above, currently, there is no technology available to study gap junction-intercellular communication in vivo. In our earlier study, we showed that M1 antibody can inhibit hemichannels, but has no effect on gap junction channel communication up to 24 hrs in cultured cells (Zhang et al., 2021). Therefore, it is unlikely this antibody will affect docking of gap junction channels given the short life of Cx43. We agreed that some phenotypes are similar between D130-136 and Cx43 cKO mice, which suggest that the importance of Cx43-forming channels. However, the lack of obvious phenotypes in R76W and the effect of M1 confirmed the major involvement of hemichannels

Figure 4 F, I, J, K and Figure 7D, G, J should be shown with values for loaded and unloaded gene expression rather than a ratio. Then a two-way ANOVA (loading and genotype) should be used to assess the effects. It is statistically unsound as shown.

We have now included values for loaded and unloaded gene expression. Student T-test is used for comparison to evaluate the response to mechanical loading of loaded and contralateral unloaded paired leg.

Figure 1 FS2 is too dark to assess. Please supply a higher contrast image.

We have provided images with higher contrast in Figure S2.

Figure 5—figure supplement 1A quantification and statistical analysis are required.

We have replaced the blot figure using two rabbit polyclonal Cx43 antibodies (Cx43CT and Cx43E2) and the mouse monoclonal Cx43M1 with MLO-Y4 membrane extracts. Because we used different secondary antibodies from different species, we cannot perform quantification and statistical analysis.

In the section titled: "PGE2 rescues impeded osteogenic responses to mechanical loading by impaired Cx43 hemichannels." It is concluded that "Together, these results demonstrate that administration of PGE2 significantly rescues impeded anabolic responses of cortical bone to mechanical loading as a result of Cx43 hemichannel inhibition." This is not supported by the data.

Our data showed that in cortical bone, the reduced bone area fraction, cortical thickness and increased bone marrow area was attenuated by PGE2 treatment. We have revised the referred section title as: PGE2 rescues impeded response of cortical bones to mechanical loading by impaired Cx43 hemichannels" and "Together, these results show that PGE2 is likely to mediate the effect of Cx43 hemichannels in response to mechanical loading".

References:

Adamo, C.T., J.M. Mailhot, A.K. Smith, and J.L. Borke. 2001. Connexin-43 expression in oral-derived human osteoblasts after transforming growth factor-β and prostaglandin E2 exposure. *The Journal of oral implantology*. 27:25-31.

Bass, S.L., L. Saxon, R.M. Daly, C.H. Turner, A.G. Robling, E. Seeman, and S. Stuckey. 2002. The effect of mechanical loading on the size and shape of bone in pre-, peri-, and postpubertal girls: a study in tennis players. *Journal of bone and mineral research : the official journal of the American Society for Bone and Mineral Research*. 17:2274-2280.

Cheng, B., Y. Kato, S. Zhao, J. Luo, E. Sprague, L.F. Bonewald, and J.X. Jiang. 2001. Prostaglandin E_2_ is essential for gap junction-mediated intercellular communication between osteocyte-like MLO-Y4 cells in response to mechanical strain. *Endocrinology*. 142:3464-3473.

Cherian, P.P., A.J. Siller-Jackson, S. Gu, X. Wang, L.F. Bonewald, E. Sprague, and J.X. Jiang. 2005. Mechanical strain opens connexin 43 hemichannels in osteocytes: a novel mechanism for the release of prostaglandin. *Molecular biology of the cell*. 16:3100-3106.

Civitelli, R., K. Ziambaras, P.M. Warlow, F. Lecanda, T. Nelson, J. Harley, N. Atal, E.C. Beyer, and T.H. Steinberg. 1998. Regulation of connexin43 expression and function by prostaglandin E2 (PGE2) and parathyroid hormone (PTH) in osteoblastic cells. *J. Cell. Biochem*. 68:8-21.

Gupta, A., S. Chatree, A.M. Buo, M.C. Moorer, and J.P. Stains. 2019. Connexin43 enhances Wnt and PGE2-dependent activation of β-catenin in osteoblasts. *Pflugers Arch*. 471:1235-1243.

Gupta, A., C. Niger, A.M. Buo, E.R. Eidelman, R.J. Chen, and J.P. Stains. 2014. Connexin43 enhances the expression of osteoarthritis-associated genes in synovial fibroblasts in culture. *BMC Musculoskelet Disord*. 15:425.

Riquelme, M.A., S. Burra, R. Kar, P.D. Lampe, and J.X. Jiang. 2015. Mitogen-activated Protein Kinase (MAPK) Activated by Prostaglandin E2 Phosphorylates Connexin 43 and Closes Osteocytic Hemichannels in Response to Continuous Flow Shear Stress. *J Biol Chem*. 290:28321-28328.

Roberts, B.C., H.M. Arredondo Carrera, S. Zanjani-Pour, M. Boudiffa, N. Wang, A. Gartland, and E. Dall'Ara. 2020. PTH(1-34) treatment and/or mechanical loading have different osteogenic effects on the trabecular and cortical bone in the ovariectomized C57BL/6 mouse. *Sci Rep*. 10:8889.

Siller-Jackson, A.J., S. Burra, S. Gu, X. Xia, L.F. Bonewald, E. Sprague, and J.X. Jiang. 2008. Adaptation of connexin 43-hemichannel prostaglandin release to mechanical loading. *J. Biol. Chem*. 283:26374-26382.

Xu, H., S. Gu, M.A. Riquelme, S. Burra, D. Callaway, H. Cheng, T. Guda, J. Schmitz, R.J. Fajardo, S.L. Werner, H. Zhao, P. Shang, M.L. Johnson, L.F. Bonewald, and J.X. Jiang. 2015. Connexin 43 channels are essential for normal bone structure and osteocyte viability. *J Bone Miner.Res.* 30:436-448.

Zhang, C., Z. Yan, A. Maknojia, M.A. Riquelme, S. Gu, G. Booher, D.J. Wallace, V. Bartanusz, A. Goswami, W. Xiong, N. Zhang, M.J. Mader, Z. An, N.L. Sayre, and J.X. Jiang. 2021. Inhibition of astrocyte hemichannel improves recovery from spinal cord injury. *JCI insight*. 6.

Zhang, Y., E.M. Paul, V. Sathyendra, A. Davison, N. Sharkey, S. Bronson, S. Srinivasan, T.S. Gross, and H.J. Donahue. 2011. Enhanced Osteoclastic Resorption and Responsiveness to Mechanical Load in Gap Junction Deficient Bone. *PLos One*. 6:e23516.